# Large-scale characterization of sex pheromone communication systems in *Drosophila*

Mohammed A. Khallaf 1,2,5✉, Rongfeng Cui 3,6, Jerrit Weißflog4, Maide Erdogmus1, Aleš Svatoš 4, Hany K. M. Dweck1,7, Dario Riccardo Valenzano 3, Bill S. Hansson1,8 & Markus Knaden 1,8✉

Insects use sex pheromones as a reproductive isolating mechanism to attract conspecifics and repel heterospecifics. Despite the profound knowledge of sex pheromones, little is known about the coevolutionary mechanisms and constraints on their production and detection. Using whole-genome sequences to infer the kinship among 99 drosophilids, we investigate how phylogenetic and chemical traits have interacted at a wide evolutionary timescale. Through a series of chemical syntheses and electrophysiological recordings, we identify 52 sex-specific compounds, many of which are detected via olfaction. Behavioral analyses reveal that many of the 43 male-specific compounds are transferred to the female during copulation and mediate female receptivity and/or male courtship inhibition. Measurement of phylogenetic signals demonstrates that sex pheromones and their cognate olfactory channels evolve rapidly and independently over evolutionary time to guarantee efficient intra- and interspecific communication systems. Our results show how sexual isolation barriers between species can be reinforced by species-specific olfactory signals.

[1] Department of Evolutionary Neuroethology, Max Planck Institute for Chemical Ecology, Jena, Germany. [2] Department of Zoology and Entomology, Faculty of Science, Assiut University, Assiut, Egypt. [3] Max Planck Institute for Biology of Ageing and CECAD Research Center at University of Cologne, Cologne, Germany. [4] Group of Mass Spectrometry and Proteomics, Max Planck Institute for Chemical Ecology, Jena, Germany. [5] Present address: Department of Neuroscience, Max Delbrück Center for Molecular Medicine, Berlin D-13122, Germany. [6] Present address: School of Ecology, Sun Yat-sen University, 135 Xingang West Road, Binjiang Road, Haizhu District, Guangdong Province, China. [7] Present address: Department of Molecular, Cellular, and Developmental Biology, Yale University, New Haven, CT, USA. [8] These authors contributed equally: Bill S. Hansson, Markus Knaden. ✉email: mkhallaf@ice.mpg.de; mknaden@ice.mpg.de

Organisms communicate with each other through exchanging signals that include visual, acoustic, tactile, and chemical (smell and taste) senses. The chemical sense is common in all organisms, from bacteria to mammals, and therefore, regarded from an evolutionary perspective as the oldest one. Animals are surrounded by a world full of odors emitted from conspecific or heterospecific individuals, as well as from the environment. The ability to exchange and decipher these signals has a significant impact on a species' success as odors help to avoid imminent threats and localize and judge food or potential mates. Olfactory systems have, therefore, evolved in a sophisticated way to meet the challenge of detecting and discriminating a countless number of odorants. While it is well established how animals use odors for intra- and interspecific communication, the evolution of olfactory systems with respect to signal production and perception is poorly understood.

One of the most crucial channels that have been suggested to contribute to speciation is the sex pheromone-sensing channels[1]. Volatile sex pheromones—airborne chemicals that stimulate sexual behaviors in the opposite sex—are the primary signals that reinforce the isolation barriers between different species. These species-specific signals often provide a full biography written in scent molecules about the sender, such as information about the reproductive and internal status. Diversification of sex pheromones among species arises via sexual, and/or natural selection[2–6]. Closely related species tend to use different pheromone blends of shared chemical compounds as a result of genetic similarities and biosynthetic pathways shared by ancestry[7–9]. This diversity in sex pheromone communication can become further affected by factors like geographical or host variations. For example, sympatric species develop pronounced divergent communication systems to overcome the risk of hybridization, while the unimpeded divergence due to geographic barriers may lead to relaxed accumulation of differences[10]. Moreover, colonization of a different host plant—an ecological adaptation—could also lead to differential sex pheromones and new ways of signal transmission and perception[11,12]. Although many studies have reported the diversity of sex pheromones among related species, the evolutionary phylogenetic history of these traits and their detection systems remains obscure.

Flies, like most animals, rely on chemical cues to locate and choose an appropriate mating partner[1,13–15]. For several reasons, flies within the genus *Drosophila* represent ideal species to study the evolution and diversity of sex pheromones, as well as their associated behaviors. First, *Drosophila* species live in an extensive range of diverse habitats across all climatic conditions, from deserts and caves to mountains and forests[16]. In these environments, drosophilids feed and breed on varied hosts such as decaying fruits, slime fluxes, mushrooms, flowers, as well as frog spawn[17]. Second, sexual behaviors of drosophilid flies differ quantitatively and qualitatively[18], which may include nuptial gift donation[19], partners' song duet[20,21], territorial dating[21], or the release of an anal fluidic droplet[22]. Third, the neural processing of pheromones in the brain of some drosophilids, especially *D. melanogaster*, is largely understood[23]. Fourth, pheromone receptors are narrowly tuned to fly odors[24] and expected to evolve at fast rates to match the dramatic diversity of pheromones among closely related species[5,25]. Lastly, out of the 52 classes of olfactory sensory neurons (OSNs) in *D. melanogaster*[26], only four respond to fly odors and are localized in a specific sensillum type[27]. Hence, the restricted number of orthologues, that are expressed in an easily identifiable and accessible sensillum type, represent promising candidates to study the coevolution of *Drosophila* pheromones and their corresponding odorant receptors (Or).

Olfactory sexual communication in *D. melanogaster* is arguably one of the best-studied systems in animals[28], and is carried out through limited chemical signals, including cis-vaccenyl acetate (*c*VA)[29]. This compound is produced exclusively by males and transferred to females during copulation, which then reduces the attractiveness of the freshly mated females[30]. Moreover, *c*VA regulates multiple behaviors: it induces sexual receptivity in virgin females[31–33], elicits aggression in males[34], modulates oviposition behaviors[35], and acts as aggregation pheromone in presence of food[36,37]. Despite the profound knowledge of *c*VA-induced behaviors in *D. melanogaster*, little is known about analogous stimuli that regulate social and sexual behaviors in other drosophilids.

Here, we identify the sex pheromones and their roles in 99 species within the family Drosophilidae, explore the evolution of pheromone signaling systems with respect to phylogenetic relationships, and highlight how sexual isolation barriers between species are reinforced by olfactory signals.

## Results

**Whole-genome information-based phylogeny of 99 drosophilids.** The genus *Drosophila* is arguably one of the most extensively studied model systems in evolutionary biology[17,38–40]. However, the phylogenetic relationships among drosophilids have suffered from low supports[41–43]. We therefore investigated the relationships of 99 species within the family Drosophilidae, 95 of which span the diversity of flies across the genus *Drosophila* (2–3 species per each (sub-)group). Whole-genome sequences (WGS) for 41 of these 99 species are available (Supplementary Data 1), thus, we generated WGS for the other 58 species (See "Methods"; accession number: PRJNA669609; available on https://doi.org/10.17617/3.5w). We, then, reconstructed the phylogeny of these 99 species using 13,433,544 amino acid sites from 11,479 genes (Fig. 1a and Supplementary Fig. 1D). Maximum likelihood (ML) phylogenetic analyses revealed strong support for the relationships among the different species (Fig. 1a). Using the four species in the Colocasiomyini subgenus as outgroups, we recovered four main clusters within the genus *Drosophila*. First, the Drosophila subgenus that contains five main groups (repleta, virilis, melanica, cardini, and immigrant groups); second, the Zaprionus subgenus that includes *Zaprionus indianus*; third, the Dorsilopha subgenus that includes *D. busckii*; fourth, the Sophophora subgenus that includes melanogaster, obscura, willistoni, and saltans groups (Fig. 1a).

**Closely related drosophilids exhibit highly divergent chemical profiles.** We then asked whether the phylogenetic relationships could reflect the differences in cuticular chemicals among these species. We, therefore, analyzed the chemical profiles of males and virgin females of all 99 species, with five replicates or more of each sex yielding more than 580 and 520 replicates, respectively (available on https://doi.org/10.17617/3.5w). Chemical analyses recognized the presence of 248 and 256 compounds (i.e., features with distinct *m/z* (mass-to-charge ratios)) across male and female replicates, respectively (Supplementary Fig. 1A, A'). Principal component analyses revealed that females of the different species groups exhibit closer distances than male species groups, indicating that females exhibit more similar chemical profiles across the species groups (Fig. 1b, b'). Similarly, in a cluster analyses of the chemical signals most males belonging to the same species group are clustered, while species groups in females are dispersed across the chemometric tree (Supplementary Fig. 1B, B'). Next, we investigate how well the males' and females' compounds agree with the phylogeny. Pairwise correlation analyses imply that indeed closely related species exhibit more similar chemical profiles in males (Fig. 1c, c'). For example, chemical profiles of male species of the *repleta* and the *melanogaster* group display

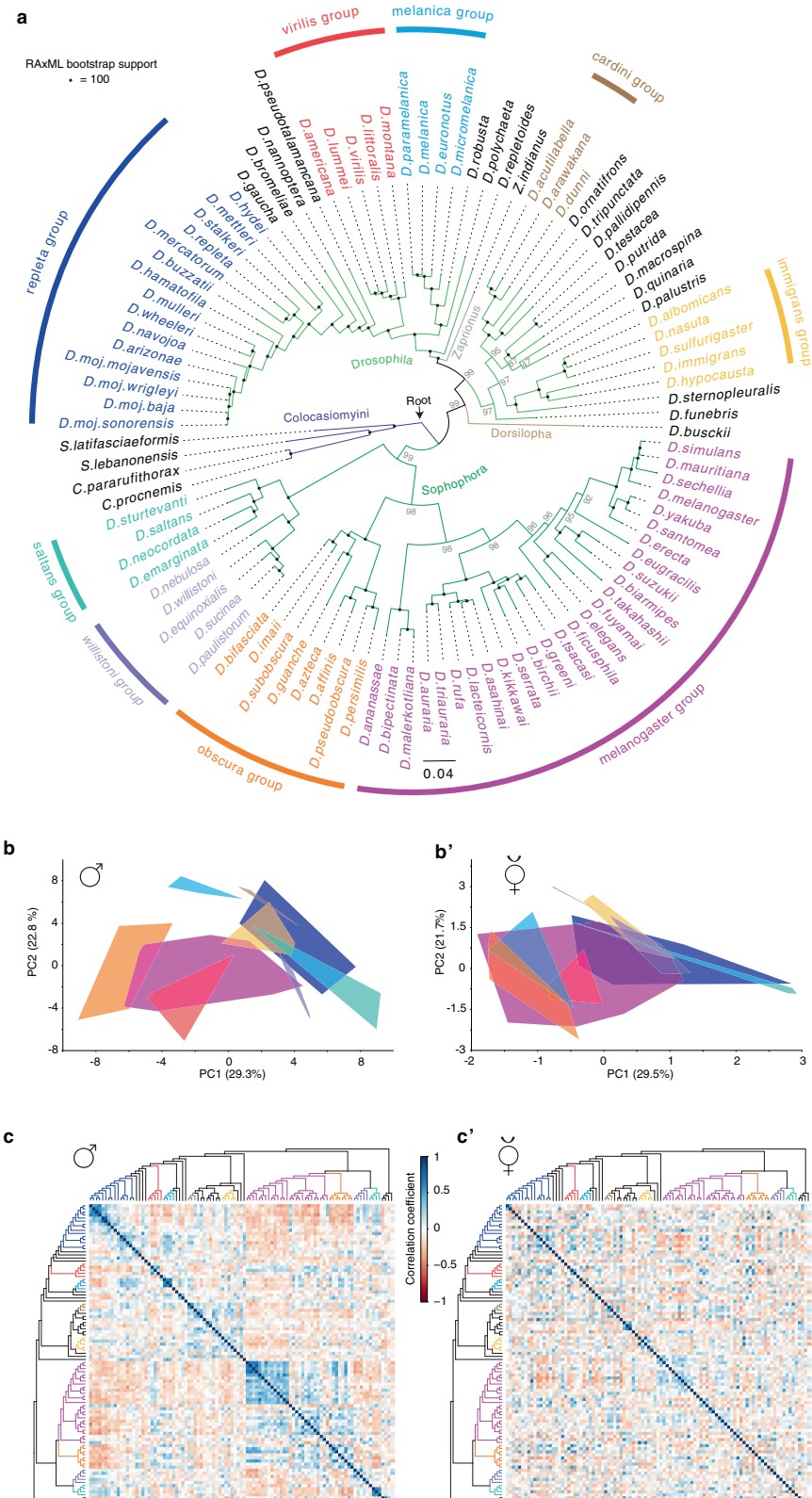

high correlation coefficients to chemical profiles of other male species of their own group, but negative correlation coefficients to chemical profiles of males of different groups (i.e., blue cells are frequently present around the diagonal) (Fig. 1c). However, female species generally display high correlation coefficients (>0.75) randomly to each other apart from their phylogenetic relationships (Fig. 1c'). Likewise, measurement of the phylogenetic signal using Pagel's λ (a measurement of the statistical dependence among species' trait values due to their phylogenetic relationships) revealed that males of related species tend to chemically resemble each other more than females ($p = 0.006$) (Supplementary Fig. 1C; Supplementary Data 2). Together, our data reveal that male chemical profiles exhibit a stronger phylogenetic signal than female chemical profiles.

**Fig. 1 Phylogenetic relationships and chemical variations among drosophilids. a** Phylogeny of 99 species within the family Drosophilidae inferred from 13,433,544 amino acids sites that represent 11,479 genes (See Supplementary Data 1 and Methods for details). Using four species in the Colocasiomyini subgenus as outgroups (purple), 95 species are distributed in four subgenera belonging to the Drosophilini tribe (Drosophila, light green branches; Zaprionus, gray branch; Dorsilopha, brown branch; Sophophora, dark green branches). Species names are color coded according to their relationships in nine different species groups, with black species depicting individual representatives of species groups. Scale bar for branch length represents the number of substitutions per site. Maximum likelihood (ML) phylogenetic analyses display strong rapid bootstrap support (100% support indicated by black circle at the nodes) for most relationships among the different species. For divergence times, see Supplementary Fig. 1D. **b** The first two principal components of male chemical profiles (data in Supplementary Fig. 1A) of the 583 replicates across the 99 male species (>5 replicates per species) based on difference in peak areas of 248 male chemical features present across these replicates (see Methods for details). Data points of each group are enclosed within the line. The lines' fill is colored according to the group identities in Fig. 1a. **b′** The first two principal components of female chemical profiles (data in Supplementary Fig. 1A′) of the 528 replicates across the 99 female species (>5 replicates per species) based on the difference in peak areas of 256 female chemical features present across these replicates. **c** Heat map showing pairwise correlations between male chemical profiles of the 99 species (ordered on each axis according to their phylogenetic relationships from Fig. 1a). Overall peak areas of 248 male chemical features across the 99 species were compared using Pearson correlation coefficient ($R^2$); Color codes in the heat map illustrate the pairwise correlations, which range from dark blue (Perfect correlation between chemical profiles) through white (no correlation) to dark red (perfect anticorrelation). The diagonal of the correlation matrix is the correlations between each species and itself (values of 1). Note that the male correlation matrix displays frequent dark blue cells mainly around the diagonal, i.e., high correlation coefficients are observed mostly between closely related species. **c′** Pairwise correlation analysis between female chemical profiles of the 99 species arranged according to their phylogeny from Fig. 1a. Overall peak areas of 256 female chemical features across the 99 species were compared using the Pearson correlation coefficient ($R^2$). See Supplementary Data 2 for the statistical Pagel's lambda correlation analysis.

**Previously unidentified potential sex pheromones undergo rapid evolution.** In drosophilids sex-specific compounds typically serve as short-range communication signals that induce or inhibit sexual behaviors[23]. For example, in the *mojavensis* complex, (*Z*)-10-heptadecen-2-yl acetate, the male-specific sex pheromone, is detected by all populations, but only induces female receptivity in the populations that produce it[22]. Similarly, in the *melanogaster* group, 7,11-heptacosadiene, a female-specific compound, induces male courtship in the producing species, but serves as an isolation barrier for the closely related non-producing species[44–46]. In search for analogous compounds all along the *Drosophila* genus, we analyzed the chemical profiles of the 99 species and compared the chromatograms of both sexes within each species (Fig. 2a). Males and females of only 18 species exhibited sexually monomorphic chemical profiles (i.e., same compounds were found in both sexes, regardless of differences in the compounds' quantity), while 81 species exhibited sexually dimorphic cuticular chemicals (a dimorphic chemical is identified as a compound that is present only in one sex) (Fig. 2b). All the 81 dimorphic species unveiled male-specific compounds (in total 43 compounds), while only 15 species exhibited female-specific ones (in total 9 compounds) (Fig. 2b; Supplementary Fig. 2A, B). Of note, most of the female-specific compounds, are long-chain unsaturated hydrocarbons (Supplementary Fig. 2A), display high boiling temperature (Supplementary Data 3), and hence are likely to be non-volatile[44]. However, male-specific compounds range between 10 to 32 carbon atoms, and belong to different chemical classes such as esters, ketones, and alkenes, as well as ether and alcohol (Fig. 2c; Supplementary Data 3). Notably, when analyzing the chemical profiles of freshly mated females, we found that many of the male-specific compounds were transferred to females during mating (green cells in Fig. 2b), reminiscent of the transfer of male-specific compounds in *D. melanogaster* and *D. mojavensis*[22,47,48]. On the contrary, none of the female-specific compounds was transferred to males during mating (Supplementary Fig. 2B).

Many of the male-specific chemicals exhibited low phylogenetic signals, and thus often are not conserved among closely related species, but sometimes present across distant species (Supplementary Fig. 2C; Supplementary Data 2). Indeed, mapping the sex-specific compounds onto the phylogenetic tree revealed that many distant species use the same male-specific compounds (Fig. 2b). However, few compounds are exclusively species- or group-specific compounds (Fig. 2b; Supplementary Fig. 2B). For example, several male-specific compounds, including *c*VA that has been thought to be restricted to the melanogaster and the immigrans groups[25], are present across several species in different groups in both subgenera Sophophora and Drosophila, while only methyl myristoleate is specific for the willistoni group (Fig. 2b). Similarly, consistent with the rapid evolution of pheromone-producing enzymes in drosophilid females[49], 7,11-heptacosadiene and 7,11-nonacosadiene, the female-specific compounds in *D. melanogaster*[23,45], are not restricted to a specific group (Supplementary Fig. 2B). This pattern supports the presence of strong selection on the sex-specific compounds to evolve fast and deviate from expectations based on stabilizing selection. In addition, our analyses revealed that 58 of the 81 dimorphic species have a blend of multiple male-specific compounds that could reach up to seven compounds, as in *D. mercatorum*, while the other 23 species employ single male-specific compounds (Fig. 2b). Overall, we identified 52 potential sex pheromones (Supplementary Data 3), which seem to evolve independently from phylogenetic constraints across drosophilids.

**Drosophilids communicate intra- and inter-specifically through rapidly evolving olfactory channels.** The volatility (i.e., low boiling points due to their shorter chain length compared to female-specific compounds; Supplementary Data 3) of most male-specific compounds suggests that they could be potential olfactory signals. We, therefore, screened for OSNs that detect male-specific compounds in *Drosophila* species via single sensillum recordings (SSR). We focused our attention on 54 species—49 dimorphic and 5 monomorphic species—because they could be successfully reared on artificial food under our lab conditions. In *D. melanogaster*, olfactory sex pheromone-responsive neurons are localized in antennal trichoid (at) sensilla, which are morphologically distinct from other sensillum types and belong to two classes (at1 and at4) that are known to be located on different antennal regions[50]. The at1 sensillum houses a single neuron (Or67d) that responds to *c*VA[32], while at4 houses 3 neurons (Or47b, Or65a/b/c, and Or88a) that respond to methyl laurate, *c*VA, and methyl palmitate, respectively[24,51,52]. Indeed, we found the at1-like and at4-like sensillum classes in all tested species except *D. pseudotalamancana* and *D. robusta*, whose at1-like sensilla could not be identified (Supplementary Fig. 3A; for identification, see Methods). We next recorded the responses of both trichoid sensillum classes in the females of 54 species to an array of chemicals (Fig. 3a), which includes 28 male-specific compounds and 8 compounds that were previously described as

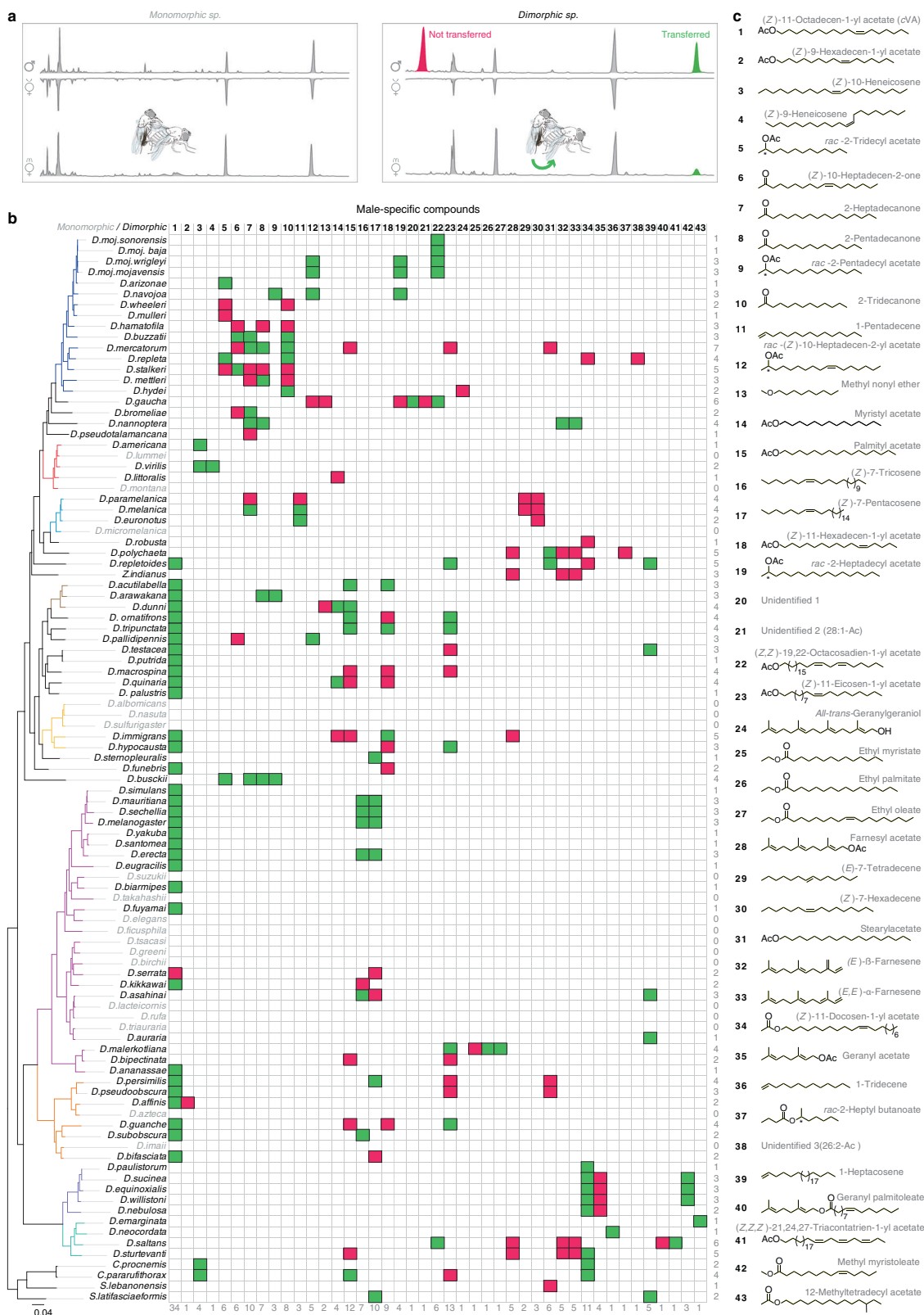

Male-specific compounds

drosophilid sex pheromones (Supplementary Data 4)[51,53,54]. The electrophysiological recording revealed that females of 36 of 49 dimorphic species detect their conspecific males' compounds (Fig. 2b) by olfactory neurons (Fig. 3b, c). Of note, flies are also able to detect many male-specific compounds of other species (Fig. 3b; Supplementary Fig. 3A, B'; Supplementary Data 5). One should note here, that our analysis focuses on the at1 and at4 sensilla that have been shown to be involved in the detection of volatile pheromones in *D. melanogaster* and close relatives[24,32,51,52], as well as in *D. mojavensis*[22]. We, however, cannot exclude that further compounds are detected by other olfactory or even gustatory sensilla types. To analyze the olfactory-based interactions between the different species, we performed network analyses, which revealed a higher olfactory

**Fig. 2 Newly identified potential sex pheromones. a** Representative gas chromatograms of virgin male (♂), and virgin (v♀), and mated (m♀) female flies obtained by solvent-free thermal desorption-gas chromatography-mass spectrometry (TD-GC-MS)[51]. Five replicates or more of each sex were analyzed, yielding more than 580, 520, and 500 replicates of males, and virgin and mated females of all 99 species, respectively. Left panel, example of a monomorphic species, whose males exhibit a chemical profile identical to that of virgin and mated females. Right panel, example of a dimorphic species that displays sexually dimorphic profiles. Colored peaks indicate male-specific compounds (green, compounds transferred to females during mating; red, non-transferred compounds). Drawings made by Mohammed A. Khallaf. **b** Distribution of 43 male-specific compounds among different drosophilids; 81 species are dimorphic species (in black), while 18 species (in grey) are monomorphic species. Phylogeny on the left side is identical to the tree in Fig. 1a; the branches are colored according to group identities. Numbers on the right side represent the sum of male-specific compounds present per species, while numbers at the bottom of the table represent number of times each male-specific compound appeared in the different species. Cell colors refer to transferred (green) and non-transferred (red) compounds. See Supplementary Fig. 2 for female-specific compounds. **c** Chemical structures and names of the male-specific compounds according to the International Union of Pure and Applied Chemistry (IUPAC). Out of 43 male-specific compounds, 40 compounds were chemically identified. Compound size ranges between 10 to 32 carbon atoms, with 23 esters, 4 ketones, 8 alkenes, 2 terpenes, 1 ether, and 1 alcohol. See Supplementary Data 3 for Kovat's Index, chemical formula, exact mass, mass spectrum (M/Z), and boiling temperature of these compounds.

clustering coefficient (i.e., the number of olfactory interactions between the species divided by a number of interactions that could possibly exist) of interspecific interactions through at1-like (for identification, see "Methods") compared to at4-like sensilla (Fig. 3d; Supplementary Fig. 3B, B'). However, self-loops, which signify the number of the intraspecific olfactory interactions (i.e., the ability of females to detect their conspecific male compounds), are comparable through at1-like and at4-like sensilla (Supplementary Fig. 3B, B'). Pairwise correlation and statistical analyses revealed that electrophysiological responses at1 and at4 neurons of the different species have low phylogenetic signals (Fig. 3e and Supplementary Fig. 3C).

We further asked whether pheromone receptors in drosophilids have evolved under positive selection. We, therefore, queried the genomic data for the orthologs of the known pheromone receptors in drosophilid flies. Of these receptors, we found in our WGS data 42, 41, and 36 orthologs of Or47b, Or67d, and Or88a, respectively, which displayed full-length sequences. We next assessed the selection pressures on these genes by computing the ratio of nonsynonymous (dN) to synonymous (dS) substitutions across the whole gene (see Methods). Statistical analyses revealed evidence of positive selection on all tested pheromone receptors with the highest pressures on the Or47b and the Or67d loci (Or47b locus, $p$-value $< 0.0001$; Or67d locus, $p$-value $< 0.0001$; Or88a locus, $p$-value $= 0.014$), indicating that pheromone receptors of the different species evolve rapidly apart from their phylogenetic relationships.

Lastly, using two different model-tuning criteria, we performed a phylogenetically corrected correlation between the evolution of male chemical phenotypes and the associated females' olfactory responses of different species and between the closely related species. Despite the high intraspecific match—females of 36 out of 49 species detect their males' compounds—(Fig. 3c), the evolution of females' responses among closely related species (limited to findings pertaining to at1 and at4 responses) does not correlate with the evolution of their male-specific compounds (Supplementary Data 6, 8–16). This implies the presence of a low interspecific correlation between detection and production. Indeed, for example, females of 19 out of 20 species, whose males produce cVA, detect this compound (i.e., cVA functions as a conspecific signal), while females of 21 out of 34 species are still able to detect cVA (Fig. 3b), although their males do not produce it (i.e., cVA functions as a heterospecific signal).

**Male-specific compounds regulate intra and interspecific sexual behaviors.** To examine the intra- and interspecific behaviors governed by the male-specific olfactory signals and to gain a better understanding of the courtship rituals of these 54 species, we recorded the sexual behaviors of conspecific couples in a single-pair courtship arena. Many species displayed different species-specific behaviors (Movies 1 to 427, available on https://doi.org/10.17617/3.5w; in total 1467 replicates, 16–48 replicates per species). For example, males of D. elegans and D. suzukii dance and spread their wings in front of females[55], D. mojavensis and D. virilis males release fluidic droplets while courting the females[22], D. subobscura males extend their proboscis to gift females with regurgitated drop of their gut contents[19], and D. nannoptera couples tend to re-mate as many as two to three times within the recording time frame of 60 min[56]. We further quantified copulation success, latency, and duration (Supplementary Fig. 4A), which varied largely among different species. Unlike the prolonged copulation time in the species of the melanogaster group, copulation lasts for <2 min in members of the repleta group (Supplementary Fig. 4A). Together, courtship recordings (available on https://doi.org/10.17617/3.5w) reveal numerous quantitative and qualitative differences in sexual behaviors among the Drosophila species.

We next focused on Drosophila species that detect their male-specific compounds via olfaction—36 out of 49 species (Figs. 3c, 4a)—and asked whether these compounds induce female receptivity. Drosophila females exhibit a preference to copulate with older males[22,51,52], which mostly possess higher amounts of male-specific compounds[22,37,57]. Therefore, we hypothesized that males perfumed with single male-specific compounds would have a higher copulation advantage than the solvent (DCM) perfumed males, which carry the same compound but a lower amount of it. In a competition-mating assay, virgin females of each species were allowed to choose between two conspecific males perfumed with a male-specific compound (Fig. 4b) or solvent [consistency of perfuming and correspondence to biologically relevant amounts were confirmed by chemical analyses; see "Methods"]. In 11 instances, females displayed a preference to copulate with males perfumed with the male-specific compound over the control ones (Fig. 4b; Supplementary Data 7). However, females of six species avoided copulating with the males perfumed with the male-specific compound (Fig. 4b).

Notably, perfuming an additional amount of cVA on males of the melanogaster clade did not increase the males' copulation success (Fig. 4b), indicating that the built-in amount of cVA in the control males is already sufficient for females' acceptance. To assure that the high copulation success of perfumed males was not due to an increased intensity of male courtship[58], we recoded their courtship activities. Courtship indices did not differ between perfumed and control males (Supplementary Fig. 4B), indicating that these compounds influence exclusively the females' sexual decisions.

Many of these olfactory-detected male-specific compounds are transferred to females during copulation (Fig. 2b; 28 out of the

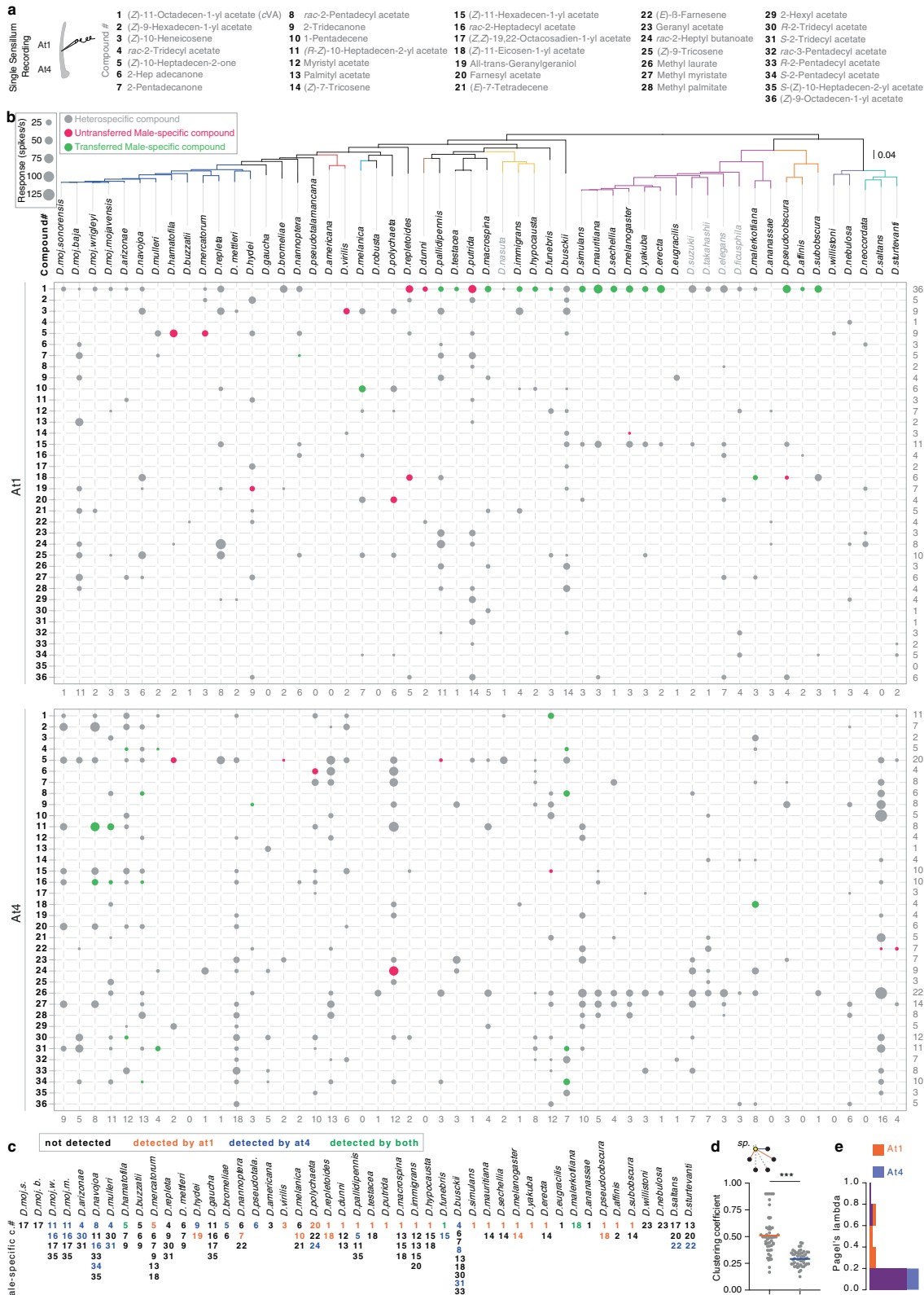

36 species in Fig. 4b). We, therefore, asked whether transfer of these compounds contributes to a general post-copulation mate-guarding strategy [as described for transferred pheromones in *D. melanogaster* and *D. mojavensis*[22,47,59,60]]. To distinguish male sexual behaviors from the female acceptance, males were offered the choice to court two headless females perfumed with a male-specific compound or solvent. We scored the first attempt to copulate with one of the rival females as a choice. However, we ensured that males do choose between perfumed females and not simply attempt copulation with the first female they court. Males in almost half of the tested species that exhibit male-transferred compounds displayed a preference to copulate with the solvent-perfumed females, including males of many species of the melanogaster group (Fig. 4c; Supplementary Data 7). Perfuming

**Fig. 3 Drosophilids communicate intra- and inter-specifically through rapidly evolving olfactory channels. a** Left: schematics of single sensillum recordings (SSR) from the antennal trichoid (at1 and at4) sensilla. Right: Names of the different chemicals used to screen the trichoid sensilla. Note that all chemicals are male-specific compounds identified in this study (Fig. 2b), except compounds# 25, 26, 27, 28, 29, 32, and 36, which were described as flies' pheromones in refs. [51, 53, 54]. **b** Color-coded electrophysiological responses towards heterospecific compounds (grey bubbles) and conspecific compounds (colored bubbles) in at1 (top) and at4 (bottom) sensilla of females of 54 species. Compound names are depicted in Fig. 3a. Red and green bubbles represent species-specific male untransferred and transferred compounds, respectively. Bubble size corresponds to the average of response values ($n =$ 3–10) ranging from 25 to 125 spikes per second. Responses less than or equal to 10 spikes per second were excluded from the bubble chart (see Supplementary Fig. 3A, A', and "Methods" for more details). Species names are arranged on the top according to their phylogenetic relationship; the tree branches are colored according to the group identities. Numbers on the right side represent the sum of species that can detect each of the male-specific compounds, while the number below the table represents the sum of chemicals that can be detected by each species. Note that the compounds' vapor pressures have no impact on the number of the olfactory responses (Supplementary Fig. 3D). **c** A summary of female's abilities to detect their own male-specific compounds through olfaction in 47 dimorphic species (two species, *D. robusta* and *D. neocordata*, whose compounds were not included among the 36 compounds, was excluded). Black numbers, undetected male-specific compounds; orange, detected by at1 neuron(s); blue, detected by at4 neurons; green, detected by both. See Supplementary Fig. 3A for more details about the number of neurons in at1 sensillum. Note that, out of 47, females of 36 species detect their conspecific male cues through at1 and/or at4. **d** Top: A schematic example of how to calculate the olfactory clustering coefficient of a given species (yellow circle) to communicate with heterospecific species (black circles) through at1 (orange lines) and at4 (blue lines). The olfactory clustering coefficient is the number of other species-specific compounds that are detected by a given species through at1 or at4 (colored lines) divided by the total number of detected and undetected species (colored + grey dashed lines). The clustering coefficient of a species is a number between 0 (i.e., no species detected) and 1 (i.e., all species detected). Below: scatter plot indicates olfactory clustering coefficients of the 54 species and their mean through at1 (orange) and at4 (blue); Two-sided Mann–Whitney $U$ test, ***$P < 0.001$ ($n = 54$ species). Note that species exhibit more olfactory intra- and inter-specific communication through at1 than at4. See Supplementary Fig. 3B, B' and "Methods" for more details on communication network analyses.
**e** Frequency histogram of Pagel's lambda estimates, which explain the correlation between the olfactory responses of at1 (orange) and at4 (blue) among the different species and their phylogenetic relationships. Note that responses of both at1 and at4 display low phylogenetic signals (i.e., do not correlate with the phylogeny). In addition, their phylogenetic signals are comparable to each other; Two-sided Mann–Whitney $U$ test, ns $P = 0.27$ between at1 and at4 responses.

with single male-specific compounds in numerous instances has no impact on the male courtship preference. This suggests that these single compounds either are not involved in the mating decision or work synergistically in combination with other compounds. On the contrary, males of *D. hydei* exhibited copulation preference for the perfumed females over the control ones, indicating that in different species male-transferred compounds can result in reverse effects. Together, male-specific compounds of 24 species regulate sexual behaviors via olfaction.

Lastly, we examined why females are still able to detect the chemical signals of heterospecific males and whether these heterospecific signals could act as reproduction isolation barriers. We focused our analysis on *c*VA due to its presence in many cosmopolitan species (e.g., *D. melanogaster*, *D. simulans*, *D. funebris*, and *D. immigrans*) that have a high chance to meet other non-*c*VA-producing species (Fig. 4d). Females of *Drosophila* species, which are able to detect *c*VA as a heterospecific signal (i.e., their males do not produce it), had the choice to mate with two conspecific males perfumed with *c*VA or solvent. Each of these females detected *c*VA with OSNs that did not detect the compounds of their conspecific males (Fig. 3b). Notably, females' preference for their conspecific males in 8 out of 13 species was significantly reduced by *c*VA (Fig. 4d; Supplementary Data 7). Indeed, perfuming males with *c*VA-like compounds—which activate the same sensillum type that *c*VA activate—resulted in comparable results (Supplementary Fig. 4C), suggesting that the activation of a *c*VA-responding neuron in this sensillum governs avoidance of heterospecific males. Overall, many male-specific compounds seem to regulate intra-sexual behaviors all along the *Drosophila* phylogeny and promote sexual isolation for heterospecific species.

## Discussion
Sexual selection imposed by the coevolution of female preferences to particular male traits leads to rapid and dramatic evolutionary divergence[61,62] and potentially contributes to speciation processes[63]. Using whole-genome sequences of 99 drosophilid species, we investigated how phylogenetic constraints impact the

evolution of cuticular hydrocarbons and potential sex pheromones per se. By linking the chemical variations and phylogenetic relationships on the one hand with the physiological responses and behavioral functions on the other, we provide large-scale evidence for the rapid coevolution of sex pheromone production and detection among drosophilid flies. The characterizations of sex pheromones, their cognate olfactory channels, and behavioral significances provide several insights into the evolution of chemical communication systems and their role in speciation.

In general, cuticular chemistry varies between closely related species in relation to their genetic relationships[64,65], geographical locations[66], and environmental factors[67]. Environmental factors are thought to have a stronger impact on the evolution of cuticular chemicals than genetic relatedness[64,68]. Our findings reveal that many of the male-specific compounds display low phylogenetic signals (i.e., less conserved in the closely related species) (Fig. 2b; Supplementary Fig. 2B, C), which might result in divergence of sexual communication among the sibling species. However, compared to females, males have significantly more chemicals with higher phylogenetic signals, i.e. there is a better correlation between genetic and chemical distance in males (Supplementary Fig. 1C). Notably, Consistent with our results, nonsexual chemical hydrocarbons in ants[64], aphids[69], ladybird beetles[70], moths[71], and drosophilids[25] exhibit gradual evolution, while aggregation pheromones in beetles display saltational (i.e., sudden and large) shifts[72]. By contrast, our observed saltational shifts in male-specific compounds contradict previous studies on the gradual mode of evolution of some of the sex pheromones in *Bactrocera*[73] and some aggregation pheromones in *Drosophila*[25]. We identified some of these previously identified aggregation pheromones as potential sex pheromones (Figs. 2b and 4b, c). The discrepancy of the mode of evolution of these sex pheromones could be explained due to binary encoding (i.e., presence or absence) of these traits among a limited number of species[25]. The saltational changes of sexual signaling are likely to occur between closely related sympatric species[74,75] to overcome the homogenizing effects of gene flow.

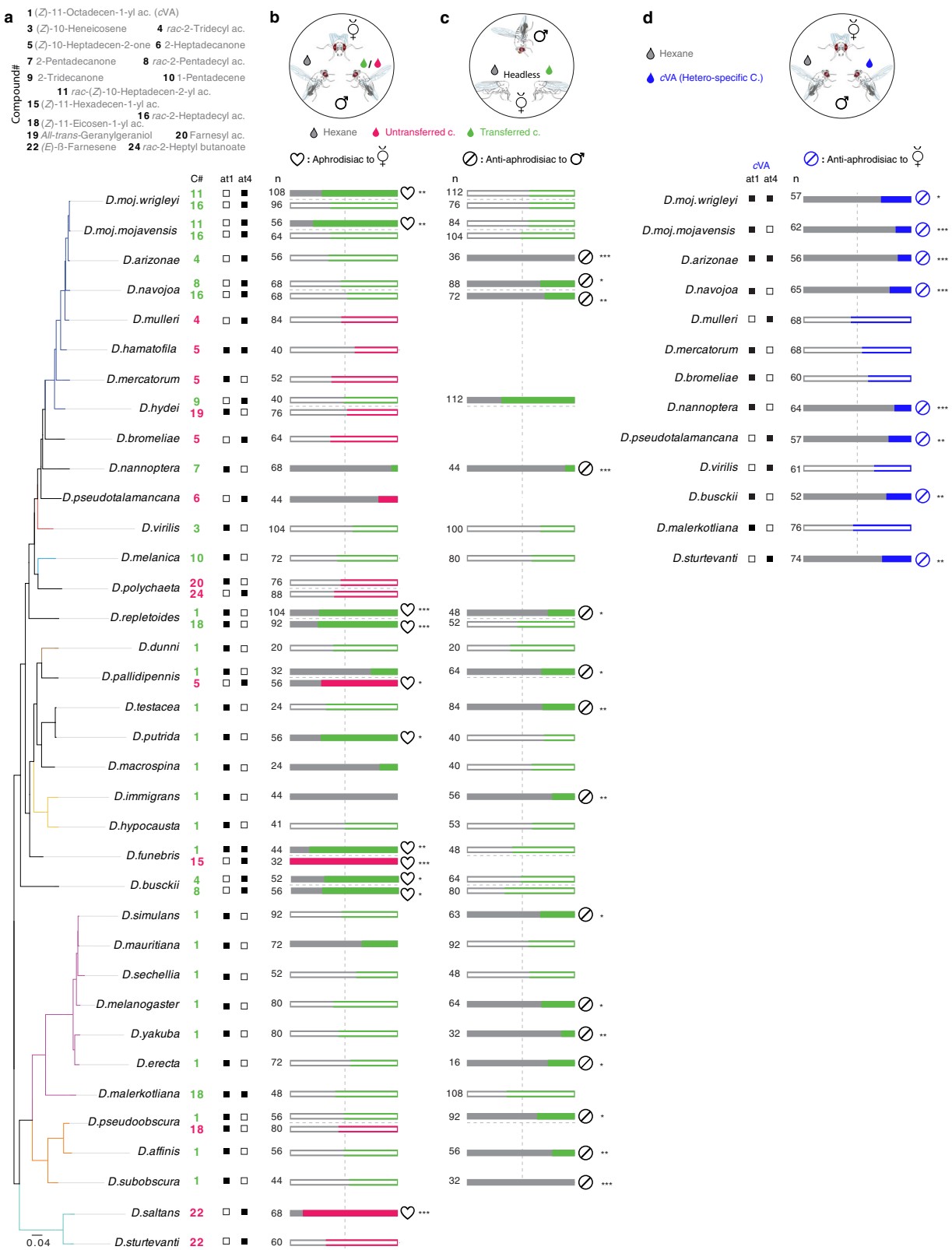

The high proportion—~82%—of species that exhibit sexually dimorphic chemical profiles (Fig. 2b) indicates the significance of chemical communication in the genus *Drosophila*. This sexual dimorphism seems to positively correlate with flies' ability to mate under low light conditions[16,76], while many of the chemically monomorphic species cannot mate in the dark[16]. The latter often display sexually dimorphic color patterns, implying that they rely on visual cues during sexual communication[76,77]. Of note, many of the sex-specific compounds exist across many species in different groups, indicating that the modification of a sexual chemical trait has occurred independently multiple times during the evolution of drosophilid flies (Fig. 2b; Supplementary Fig. 2B). For example, *c*VA, (*Z*)-11-eicosen-1-yl acetate, and palmityl acetate are present in 34, 13, and 12 species that belong

**Fig. 4 Male-specific compounds regulate intraspecific sexual behaviors and interspecific sexual isolation. a** Top left: Names of the compounds that are exclusively produced by males of 54 species (left below) and detected by conspecific females through at1 or/and at4 (right below). These compounds were used for the behavioral experiments in Fig. 4b–d. **b** Top: Schematic of a mating arena where females of each species had the choice to mate with two conspecific males perfumed with their olfactory-detected male-specific compound (indicated, in Fig. 4a, on the left side of the horizontal dashed line) or solvent (dichloromethane, DCM). For consistency of perfuming and correspondence to biologically relevant amounts see "Methods". Below: bar plots represent the percentages of copulation success of the rival males. Results from females that were only courted by one male were excluded. In this and other panels, filled bars indicate a significant difference between the tested groups; ns $P > 0.05$; $*P < 0.05$; $**P < 0.01$; $***P < 0.001$, chi-square test. Number of replicates are stated on the left side on the bar plots. See Supplementary Fig. 4A for details regarding the differences and similarities of sexual behaviors among the 54 species. Note that in 11 instances, females displayed a preference to copulate with the male-specific compound-perfumed males over the control ones, while 6 compounds resulted in avoidance, and 29 turned out to be neutral. See Supplementary Data 7 for raw data and statistical analyses. See Supplementary Fig. 4B for the effect of perfuming on the males' courtship behavior. Drawings made by Mohammed A. Khallaf. **c** Top: Competition courtship arenas where a male of each species had the choice to court two decapitated conspecific females perfumed with the male-transferred compound. Note that we tested only transferred compounds (green). Below: bar plots represent the percentage of the first copulation attempts towards perfumed and control females. Results from males that only courted one female were excluded; see Methods. Note that 15 compounds inhibited courtship, 1 compound increased courtship and 16 compounds turned out to be neutral. Drawings made by Mohammed A. Khallaf. **d** Top: Schematic of a mating arena where a female of each species had the choice to mate with two conspecific males perfumed with olfactory-detected heterospecific cVA or solvent (DCM). Note that we only tested the species that do not produce but still detect cVA. Below: bar plots represent the percentages of copulation success of the rival males. Results from females that were only courted by one male were excluded. Drawings made by Mohammed A. Khallaf.

to different groups, respectively. Instead, the production of 2-heptadecanone and (Z)-7-pentacosene is present in different groups but restricted to higher taxonomic levels (i.e., subgenre Drosophila and Sophophora, respectively). Moreover, 14 compounds have appeared on only one occasion across the 99 species. The observed saltational changes are not necessarily unexpected as minor mutations suffice to induce large-scale changes in the biosynthetic pathways of sex pheromones[71,78,79]. Likewise, gene families involved in biosynthesis of cuticular chemicals have been shown to evolve rapidly and independently among closely related drosophilids[80].

One key observation of our study is the diversity and abundance of male-specific compounds compared to female ones—43 compared to 9, respectively—across the dimorphic species (Fig. 2b; Supplementary Fig. 2B). Surprisingly, 81 dimorphic species exhibit male-specific compounds, while only 15 species have female-specific compounds (Fig. 2b; Supplementary Fig. 2B). This could be attributed to the fact that drosophilid females are regarded as "the choosy sex", which rely on volatile male sex pheromones to find a high-quality conspecific male[32,36,53,81] and to avoid costly interspecific mating[22,32,82]. Moreover, males are found to court heterospecific females in equal vigor as conspecific females[22,83,84], even after learning the conspecific females' chemical profiles[85]. Similarly, males exhibit a higher preference for females that exhibit no cuticular hydrocarbons (i.e., females lacking oenocytes (" oe−" females)) over wild-type females, while females are less receptive to oe− males[86]. Furthermore, male cuticular hydrocarbons are modulated more easily by lab-induced natural and sexual selection than female cuticular hydrocarbons[87]. All these reasons, aside from the females' strong preferences for male sexual traits, seem to have resulted in stronger selection pressures on the cuticular hydrocarbons of drosophilid males.

To match the diverse male chemical traits, females are expected to coevolve cognate sensory detection systems to permit mate recognition[1]. Drosophilid chemoreceptor genes evolve rapidly[88] and single point mutations can result in species-specific variance of receptor tuning[89]. Such specificity has shown to be not random and principally occurred to match chemical divergence associated to host selection[88–92] or mate recognition[93]. Similarly, we found that pheromone-responsive olfactory channels of at1 and at4 sensilla evolve high selectivity that permits an extreme fit to the evolution of sex pheromones in conspecific partners (Fig. 3b). Females of more than 75% of the dimorphic species are able to detect their diverse conspecific male-specific compounds through

the same olfactory channels (at1 and at4 neurons) (Fig. 3b), suggesting that their cognate olfactory receptors are under positive selection that has acted strongly to modify their functional capabilities. Similar to the evolution of male-specific compounds, the functional divergence of these olfactory channels among the closely related drosophilids is not correlated with their phylogeny nor with the evolutionary changes in male chemical profiles (Fig. 3e, Supplementary Data 6, and ref. [90]). Moreover, we found that many species detect other heterospecific male compounds, highlighting the broad potential for interspecific olfactory communication among the different drosophilids (Fig. 3b). Behavioral experiments revealed that heterospecific signals reduce the likelihood of hybridization through different olfactory channels from those specialized to detect conspecific pheromones (Fig. 3b). For example, the subspecies of D. mojavensis, as well as D. arizonae and D. navojoa detect their own pheromones through their at4-like sensilla, while they detect the heterospecific cVA through the at1-like sensilla (Fig. 4d). Interestingly, contrary to that, at1-sensilla are used in D. melanogaster to detect conspecific pheromones[32]. These results reveal that species retain—at the peripheral level—the ability to detect the chemicals no longer produced by conspecifics, but a change in valence is likely encoded at the level of central circuits[94]. In line with our findings, previous studies have shown that heterospecific sex pheromones could reinforce the sexual isolation among sympatric species or recently diverged populations through conserved peripheral olfactory pathways[22,45].

Unlike cVA-induced behaviors in D. melanogaster, which are encoded mainly through a single olfactory channel[32], sexual behaviors of many other drosophilids seem to be mediated by different compounds through multiple channels (Figs. 3b, 4b). The lack of genetic tools for most of the drosophilid species currently precludes further investigations of the genetic and neuronal correlates of intraspecific sexual behaviors and interspecific sexual isolation. A future challenge will be to investigate the genetic basis of the rapid evolutionary rate of sex pheromone production and detection and how these chemicals, together with the other sensory signals, collaborate to result in the birth of new species.

## Methods
### Drosophila lines and chemicals
*Fly stocks.* Wild-type flies used in this study were obtained from the National *Drosophila* Species Stock Centre (NDSSC; http://blogs.cornell.edu/drosophila/) and Kyoto stock center (Kyoto DGGR; https://kyotofly.kit.jp/cgi-bin/stocks/index.cgi).

Stock numbers and breeding diets are listed in Supplementary Data 1. All flies were reared at 25 °C, 12 h Light:12 h Dark and 50% relative humidity. For more details on the food recipes see *Drosophila* Species Stock Centre (http://blogs.cornell.edu/drosophila/recipes/). Care and treatment of all flies complied with all relevant ethical regulations.

*Chemicals.* Male- and female-specific compounds are listed in Supplementary Data 3, while compounds used for SSR and behavior, their sources and CAS numbers are listed in Supplementary Data 4. All odors were diluted in dichloromethane (DCM) for SSR and behavioral experiments.

## Whole-genome sequencing and phylogenetics

*Sequencing library preparation.* Genomic DNA was extracted from a single fly per each species (for more details see Supplementary Data 1) using qiagen DNeasy blood and tissue kit (cat. nos. 69504). Extracted DNA (~20 ng/µl) was quantified with Qubit broad range dsDNA kit, and diluted to a concentration of 1 ng/µL. Tagmentation was performed with in-house Tn5 transposase prepared with a previously described method (Picelli et al. [95]; Genome Research). Tagmented fragments were purified with 1 volume of SPRI beads (1 mL SeraMag GE Healthcare, 65152105050250 beads in 100 mL of PEG8000 20%, NaCl 2.5 M, Tris-HCl pH = 8.0 10 mM, EDTA 1 mM, Tween20 0.05%), and subjected to 20 cycles of Kapa HiFi PCR enrichment with barcoded primers using the following cycling conditions: 72 °C 3 min, 98 °C 1 min, 20 cycles of 98 °C 45 s, 65 °C 30 s, 72 °C 30 s. An equal volume of PCR products was pooled and purified with SPRI beads with a two-sided size selection protocol, using 0.55X (of the PCR pool volume) SPRI beads for the first selection and 0.2X SPRI beads for the second. Library pool was quantified with Qubit broad range dsDNA kit and sized with TapeStation D1000. Sequencing was performed on two HiSeq X lanes. Genomes are available on NCBI with accession number: PRJNA669609.

*Gene annotations and determination of orthologs.* Nine draft assemblies deposited on NCBI genbank (*D. albomicans*, *D. americana*, *D. montana*, *D. nasuta*, *D. pseudoobscura*, *D. robusta*, *D. subobscura*, *S. lebanonensis*, *P. variegata*) were not annotated. We lifted over annotation information from *Drosophila melanogaster* for these genomes by performing blast, followed by exonerate and genewise alignments as previously described. We classified annotated genes by clustering protein-coding sequences from 31 species using UPhO as previously described. Together, 11575 orthologs were identified from the annotated genomes. Together with already annotated genomes (*n* = 22), they serve as reference genomes to which short reads from other species were mapped.

ORs were identified from the UPhO ortholog assignment pipeline by requiring the ortholog to include an annotated *D. melanogaster* OR. The gene name of an ortholog is then assigned by the *D. melanogaster* gene name. For re-sequenced species, coverage filters were applied as described for the genes used for phylogenetics. Genes with excessive coverage in re-sequenced species were completely discarded.

*Read processing and generation of pseudogenome assemblies.* Raw reads were demultiplexed with dual barcodes by the sequencing facility, and trimmed to remove any adapter sequences using Trimmomatic version 0.32 using the following parameters: ILLUMINACLIP:illumina-adaptors.fa:3:7:7:1:true LEADING:25 TRAILING:25 SLIDINGWINDOW:4:20 MINLEN:50. We next determined the optimal reference genome to use by mapping the first 10,000 paired-reads with BWA-MEM to each of the 31 reference genomes, followed by computing the proportion of properly mapped read pairs $Pproper$ and the averaged mapping quality $MAPQ$. We designed an ad hoc index to maximize data usage, reference quality index = (completeness of reference genome annotation) * $Pproper$ * (1-10^$MAPQ$). The reference with the highest reference quality index was chosen for each short-read dataset. Pseudogenomes were produced as previously described, by mapping reads to the best reference, realigning around gaps, and substituting bases of the reference genome and masking regions with no mapped reads ($MAPQ < 20$).

*Alignment and phylogenetics.* Orthologous protein-coding sequences were extracted from reference genomes and pseudogenomes by using the GFF annotations of the corresponding reference species. TranslatorX was used to align the coding-sequencings by codon, and cleaned with GBlocks (MinSeqConsv = 0.5, MinSeqFlank = 0.55). Aligned protein-coding sequences were concatenated for each species, resulting in the final alignment matrix with 11,479 genes, 13,433,544 sites in 99 species (5 samples were excluded based on a preliminary tree, due to their clear contradiction with well-established taxonomy, suggesting potential problems in mislabeling or strain contamination). Data completeness ranges from 4.46–97.27% (mean = 58.59%). Partitioning the full alignment into 3 codon positions, we inferred a maximum likelihood tree by using RAxML 8.2.4 with 100 rapid bootstrap supports. Because branch length may not be accurate with extensive missing data, we then further optimized the branch lengths with ForeSeqs using a branch-length stealing algorithm using the parameters "–branches s–threshold 0.5". Due to computational constraints, only the top 500 most informative genes were used to re-optimize branch lengths.

## Chemical analyses

*Thermal desorption-gas chromatography-mass spectrometry (TD-GC-MS).* Individual headless male and female flies in different mating status (virgin or freshly mated (within 1 h from the successful mating)) were prepared for chemical profile collection as described previously[51], with some modifications. Briefly, the GC-MS device (Agilent GC 7890 A fitted with an MS 5975 C inert XL MSD unit; www.agilent.com) was equipped with an HP5-MS UI column (19091S-433UI; Agilent Technologies). After desorption at 250 °C for 3 min, the volatiles were trapped at −50 °C using liquid nitrogen for cooling. In order to transfer the components to the GC column, the vaporizer injector was heated gradually to 270 °C (12 °C/s) and held for 5 min. The temperature of the GC oven was held at 50 °C for 3 min, gradually increased (15 °C/min) to 250 °C and held for 3 min, and then to 280 °C (20 °C/min) and held for 30 min. For MS, the transfer line, source, and quad were held at 260 °C, 230 °C, and 150 °C, respectively. Eluted compounds for this and the following analyses were ionized in electron ionization (EI) source using electron beam operating at 70 eV energy and their mass spectra were recorded in positive ion mode in the range from *m/z* 33 to 500. The structures of the newly identified compounds were confirmed by comparing their mass spectra and retention times of the synthesized or commercially available standards (for more details see Supplementary Data 4. The age of males and females is 10 days.

*Body extract analysis by GC-MS.* Fly body extracts were obtained by washing single flies of the respective sex and mating status in 10 µl of hexane for 30 min. For GC stimulation, 1 µl of the odor sample was injected in a DB5 column (Agilent Technologies; www.agilent.com), fitted in an Agilent 6890 gas chromatograph, and operated as described previously[96]. The inlet temperature was set to 250 °C. The temperature of the GC oven was held at 50 °C for 2 min, increased gradually (15 °C/min) to 250 °C, which was held for 3 min, and then to 280 °C (20 °C/min) and held for 30 min. The MS transfer-line, source, and quad were held at 280 °C, 230 °C, and 150 °C, respectively. XCMS[97]—a bioinformatics software (version 3.7.1) designed for statistical analysis of mass spectrometry data—was used to analyze the chemical profiles of males and females of the different species.

*Chiral chromatography.* To check the presence of different stereoisomers of some compounds, hexane body extracts of male flies were injected into a CycloSil B column (112–6632, Agilent Technologies; www.agilent.com) fitted in Agilent 6890 gas chromatograph and operated as follows: The temperature of the GC oven was held at 40 °C for 2 min and then increased gradually (10 °C/min) to 170 °C, then to 200 °C (1 °C/min), and finally to 230 °C (15 °C/min) which was held for 3 min. All gas-chromatography data were collected by MSD Chemstation software (F.01.03.2357).

*Perfuming flies with male-specific compounds.* Male and female flies were perfumed with the compounds singly diluted in DCM or DCM alone as previously described[22]. Briefly, 10 µL of a 50 ng/µL stock solution was pipetted into a 1.5-mL glass vial. After evaporating the DCM under nitrogen gas flow, ten flies were transferred to the vial and subjected to three medium vortex pulses lasting for 30 s, with a 30-s pause between each pulse. Flies were transferred to fresh food to recover for 2 h and then introduced to the courtship arenas or subjected to GC-MS analysis to confirm the increased amount of the perfumed acetate. Each fly was coated with ~2–10 ng of the compound of interest.

## Chemical identification and synthesis. (See the Supplementary Information File).

## Behavioral experiments

*Single and competitive mating assays.* Males and females were collected after eclosion and raised individually and in groups, respectively. Single-pair courtships assays were performed in a chamber (1 cm diameter × 0.5 cm depth) covered with a plastic slide. Courtship behaviors were recorded for 60 min using a GoPro Camera 4 or Logitech C615 as stated in the figure legends. All single mating experiments were performed under normal white light at 25 °C and 70% humidity. Each video was analyzed manually for copulation success, which was measured by the percentage of males that copulated successfully, copulation latency, which was measured as the time taken by each male until the onset of copulation, and copulation duration. The competition courtship experiments (competitive experiments with two males and one female; competitive experiments with two females and one male) were performed in a chamber (5 cm diameter × 1 cm depth). In all competition experiments, copulation success was manually monitored for 1 h. Decapitated females were used in the courtship assays to disentangle male sexual behaviors from female acceptance.

In the competition mating assays, rival flies were marked by UV-fluorescent powder of different colors (red: UVXPBR; yellow: UVXPBB; green: UVXPBY; purchased from Maxmax.com; https://maxmax.com) 24 h before the experiments. Competition assays were manually observed for 1 h and copulation success was scored identifying the successful rival under UV light. Decapitated females were used to observe the first copulation attempt of males in presence of the different compounds and DCM perfumed conspecific females. Data from competition experiments represents either female courted by both rival males or males courted with both rival females to ensure that females or males chose between rival pairs

and did not simply copulate or court with the first partner they encountered. Results from females that were only courted by one male, or males that only courted one female were excluded. All courtship and copulation data were acquired by a researcher blind to the treatment.

### Electrophysiological experiments

*Single sensillum recording (SSR).* Female flies were immobilized in pipette tips, and the third antennal segment was placed in a stable position onto a glass coverslip[98]. Trichoid sensilla were identified based on their sensillum morphology under a microscope (BX51WI; Olympus) at ×100 magnification. The two different classes of trichoid sensilla were identified on the basis of their anatomical location (at1 sensilla in the central region, while at4 sensilla in the distolateral region of the antenna) and spontaneous activities (at1 sensilla house less neurons than at4 sensilla), which are known from *D. melanogaster*[50]. The extracellular signals originating from the OSNs were measured by inserting a tungsten wire electrode in the base of a sensillum and a reference electrode into the eye. Signals were amplified (Syntech Universal AC/DC Probe; Syntech), sampled (10,667.0 samples/s), and filtered (300–3000 Hz with 50/60 Hz suppression) via USB-IDAC connection to a computer (Syntech). Action potentials were extracted using AutoSpike software, version 3.7 (Syntech). Synthetic compounds were diluted in dichloromethane, DCM, (Sigma-Aldrich, Steinheim, Germany). Prior to each experiment, 10 μl of the diluted odor was freshly loaded onto a small piece of filter paper (1 cm$^2$, Whatman, Dassel, Germany), and placed inside a glass Pasteur pipette. Similar to ref. [22], our preliminary electrophysiological recordings revealed that high concentrations of odorants (e.g., $10^{-1}$ dilution (v/v)) elicit strong responses that might saturate or kill the olfactory neurons, while low concentration of odorants (e.g., $10^{-5}$ dilution (v/v)) elicit no or low responses. Therefore, an intermediate concentration ($10^{-3}$) has been used for all odorants. The odorant was delivered by placing the tip of the pipette a few millimeters away from the antennae, to ensure the delivery of the low volatile chemicals[99]. Neuron activities were recorded for 10 s, starting 2 s before a stimulation period of 0.5 s. Responses from individual neurons were calculated as the increase (or decrease) in the action potential frequency (spikes/s) relative to the pre-stimulus frequency. Traces were processed by sorting spike amplitudes in AutoSpike, analysis in Excel and illustration in Adobe Illustrator CS (Adobe systems, San Jose, CA). Note that number of neurons per same sensillum type is not conserved in the different *Drosophila* species—as revealed by number of at1 neurons across the different species in Supplementary Fig. 3A. Moreover, sorting the number of neurons based on the spike amplitudes in all at4 and some at1 sensilla is technically challenging due to the close spike amplitudes of the sensillum neurons.

### Statistical analyses

*Estimating phylogenetic signal with Pagel's λ.* Raw peak signals were first standardized by dividing the area under each peak by the sum of areas under all peaks. For each sex, the corresponding peaks were aligned, and the standardized signals across samples were logarithm-transformed to approximate normality, followed by standardization with a z-transformation. The phylogenetic signals contained in each chemical component were estimated by combining the transformed peak intensity with the DNA phylogeny, using the phylosig function in the phytools R package (Version 1.1.447). We compared the distribution of Pagel's λ between sexes using the unpaired Wilcoxon rank sum test. In order to test whether correlations exist between chemical production and neuronal responses, we applied phylogenetic generalized linear models (PGLS). Raw neuronal response values were used as independent variables, and only z-transformed because statistical test (Shapiro-Wilks test) revealed normal distribution of the dataset. Chemical levels were transformed as described in the previous section, and two encoding methods were used for the chemical levels—binary for presence or absence, or continuous. When the chemical levels were binary-encoded, we used phylogenetic logistic regression implemented in the R package phylolm and 2000 bootstraps to determine statistical significance. For continuous encoding of the chemical levels, we used the PGLS method implemented in the R package caper, with the optimal branch transformation model determined by model selection with BIC as previously described[100].

### Selection pressure analysis

BUSTED (Branch-Site Unrestricted Statistical Test for Episodic Diversification) was used to assess if a gene has experienced a positive selection at any site at the gene-wide level. BUSTED approach is available at the datamonkey web server (https://www.datamonkey.org/)[101]. All branches of the three phylogenetic trees—including 42, 41, and 36 orthologs of *Or47b*, *Or67d*, and *Or88a*, respectively—were entirely tested for positive selection.

### Statistics and figure preparations

The normality test was first assessed on datasets using a Shapiro test. Statistical analyses (see the corresponding legends of each figure) and preliminary figures were conducted using GraphPad Prism v. 8 (https://www.graphpad.com). Figures were then processed with Adobe Illustrator CS5.

**Reporting summary**. Further information on research design is available in the Nature Research Reporting Summary linked to this article.

### Data availability

All relevant data supporting the findings of this study and all unique biological materials generated in this study are available https://doi.org/10.17617/3.5w. The whole-genome sequences are available via the accession code PRJNA669609.

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

## Acknowledgements

We thank Ibrahim Alali for fly rearing and helping in the courtship assays, and Maide Erdogmus for helping in the electrophysiological recordings. We are grateful to Kerstin Weniger, Sybille Lorenz, Silke Trautheim, Carolin Hoyer, and Lisa Merkel for their technical support. Many thanks to Richard Benton and Rowan French for discussions and comments on the manuscript, and to Marcus Stensmyr for discussions on the phylogeny. Wild-type flies were obtained from the San Diego Drosophila Species Stock Center (now The National Drosophila Species Stock Center, Cornell University) and KYOTO Stock Center. This research was supported through funding by the Max Planck Society.

## Author contributions

M.A.K., H.K.M.D., B.S.H. and M.K. conceived the project. All authors contributed to the experimental design, analysis, and interpretation of results. M.A.K. prepared all figures and collected all experimental data. R.C. and D.R.V. reconstructed the phylogeny and performed all phylogenetic analyses. A.S. identified and J.W. synthesized the sex-specific compounds. Other experimental contributions were as follows: M.A.K. (Figs. 1, 2, 3a–d, 4, Supplementary Figs. 1B-B', 2, 3, 4, movies 1–427, Supplementary Data 1, 3–5, and 7, and Supplementary Data 8–16), R.C. (Fig. 1a, 3e, Supplementary Figs. 1A, A', and C, D, 2C, and Supplementary Data 2, 4), J.W. (Fig. 2c, and Supplementary Fig. 2A). M.A.K. wrote the original manuscript, and M.K., R.C., D.R.V. and B.S.H. contributed to the final manuscript. All coauthors contributed to the subsequent revisions.

## Funding

## Competing interests

The authors declare no competing interests.
