## [Peer Review File · Nature Communications]

REVIEWER COMMENTS

Reviewer #1 (Remarks to the Author):

The manuscript by Khallaf et al represents an impressive dataset and resource examining pheromone communications among 99 drosophilid species. The authors present massive data examining the phylogenetic relationship among 99 drosophila species, the chemical profiles of these species (as detected by GC), which of these chemicals are species specific, and which are transferred during mating. The authors also examine mating behaviors of 54 species that displayed sexually dimorphic pheromone profiles transferred during mating. Finally, the authors record from the two trichoid sensilla in 54 species to examine the extent to which chemicals are detected by the olfactory systems. Among the many findings, the authors present evidence for many new sexually dimorphic pheromones, male specific pheromones in additional species, the independent evolution of chemical profiles and the olfactory system's ability to detect these chemicals, and multiple strategies for speciation among closely related species. The data presented here provide a solid foundation for the extensive investigation of chemical communication between species and will remain a valuable resource for many years to come.

I have no major concerns regarding this work, and strongly support its publication. At times the manuscript could help clarify terms that might be unfamiliar to a more general audience. Such as a description of Pagel's λ , 'low phylogenetic signal' (line 187), etc. Also, please add to the figure legend for Figure 3 when male or female at1/at3 are being recorded. The authors may wish to consider uploading their mating movies to Youtube or another service, which can be an effective method of storage and distribution.

Reviewer #2 (Remarks to the Author):

Khallaf et al examine the evolution of sex pheromones, olfactory detection pathways, and sexual behavior among up to 99 species of Drosophilid flies. Drosophila provides a remarkable system for addressing these phenomena, with species showing diverse ecologies and sexual signals despite relatively well known receptors and olfactory circuits. They show that even closely related species often differ in sexual signaling and behavior and test for levels of phylogenetic constraint in various features of the system.

The work that went into this study, and the richness of the data produced, are truly impressive – chemical analysis of 5 males and 5 females from each of 99 species! Extracellular recordings from female antennal sensilla in ~50 species! Mating behavioral assays in ~50 species! These data will provide a huge resource for the Drosophila community.

The analyses performed are less strong, in my opinion, than the data themselves. The questions and hypotheses are often ambiguous, making it difficult to compare specific results to expected outcomes

and interpret accordingly. As a result, interpretations sometimes felt arbitrary, surface-level, and/or poorly supported by the data.

Overall, I think the data are important and the work is potentially publishable, but I recommend a significant revision, involving (1) clarification of hypotheses and expected outcomes and (2) more careful and measured interpretations. I would also like to see (3) new, more incisive analyses that are a better match for this amazing dataset and the newly clarified hypotheses/predictions.

Technical issues

I think the study is technically sound. My only substantial comment is that vapor-phase concentrations of the stimuli used for the single sensillum recordings are not clear. Were vapor-phase concentrations of delivered stimuli directly measured to ensure biological relevance? If not, how do the authors know that the concentrations are appropriate? And how does uncertainty in this regard affect conclusions? It seems to me that the most important issue is whether some fraction of biologically relevant olfactory responses may have been missed due to unnaturally low vapor-phase concentrations (e.g. for compounds with limited volatility). The various circle sizes in Figure 3b also give the impression that quantitative differences in the response of a given sensillum to different compound reflect neural sensitivity, when they may instead reflect differences in compound volatility (and consequent variation in vapor-phase concentration).

Conceptual issues

My most significant critique of the study has to do with the conceptual foundation and interpretations. The authors frame the study as a test of rapid coevolution between sex pheromones and their cognate sensory channels, but it's not clear to me exactly which analysis tests for coevolution. We learn about the degree to which various aspects of species and sex specific signals are phylogenetically constrained. We gain a ton of valuable information about what pheromones are produced, and what neurons may be used to detect them, and how they may affect mating behavior in a staggering number of species. But it is not totally clear what the key hypotheses are (eg. coevolution vs no coevolution? rapid vs slow evolution?...). The analyses are largely descriptive – rather than testing specific, well-defined hypotheses and predicted outcomes. If the hypotheses pertain to evolutionary rates and phylogenetic constraint, then what serves as the reference point (i.e. neutral expectation) for saying that something is evolving rapidly? Or that something is coevolving?

The first analysis of chemical signals comes closer to testing specific hypotheses than other analyses in the paper. The authors conduct an interesting comparison between the rates of evolution of male vs. female compounds, and, within male compounds, of male-specific versus shared compounds. However, the interpretations are internally inconsistent. When comparing male signals to female signals, the reduced level of phylogenetic constraint for female compounds (Supp fig 1d) is interpreted as showing that they are less important for species-specific behaviors. However, when comparing male-specific compounds to other compounds found on males, the reduced level of phylogenetic constraint for male-

specific compounds is interpreted as rapid evolution via positive selection and saltational shifts. Why the different interpretations of the same type of comparative signal? I recommend conducting a single, more wholistic analysis of three types of compounds... female-specific, male-specific, and shared. The distribution of Pagel's lambda for these three could be shown together in the same plot and discussed at the same time. Alternative explanations should be considered. For example, decreased Pagel's lambda (reduced phylogenetic constraint) could result from both increased positive selection or decreased stabilizing selection. The authors should also consider other analyses that may allow them to distinguish among these hypotheses (see some suggestions below).

Another place where the data are extremely rich, but it is not clear to me what the prediction is and, whether it has been upheld, is in the discussion/analysis of detection of pheromones by at1 and at4 sensilla. Fig. 3b represents an impressive amount of work. So much information is contained there. The authors note that females of more than 75% of species can detect compounds on conspecific males through these sensilla – suggesting positive selection. Where is the test of positive selection here? Is it Pagel's lambda? What value of Pagel's lambda do we expect to see if there is no positive selection? They then go on to show that females of many species also detect compounds found on heterospecific males through these same sensilla – discussing the idea of evolutionary labile central circuits toggling the behavioral valence of conserved peripheral signals. A single species is given as an example, and a very interesting anecdote, but I can't find a broad test of this hypothesis anywhere in the analysis. For example, when closely related species diverge in male-compounds, how often is this accompanied by a change in the female sensillum that detects those compounds... and/or by a shift in behavioral valence? Is the ability of females to detect heterospecific compounds largely driven by retention of detection ability for compounds that are produced by other species in the same clade? Or is detection ability more widely dispersed across *Drosophila* phylogeny – and, perhaps, better explained by sympatry than phylogeny?

Finally, I was confused about the hypothesis being tested by the female-choice mating assays described in Fig 4b (and lines 284-300). What is achieved by adding more of a male-specific compound to a male that already has this compound? And then testing him against an unmanipulated male that also presumably has this compound? Is this a test of whether quantitative levels of male-specific compounds affect mating? And what is the prediction? Should we expect elevated levels of these compounds to be more attractive to females than their natural levels? It is not clear what these experiments are telling us about fly biology. Unless the authors can significantly clarify the predictions/expected outcomes/interpretations, I suggest removing these experiments from the study and focusing on the more-interpretable male-choice experiments (Fig. 4c). However, because SSR was performed only on females (as far as I can tell), the authors may need to find another way to justify the choice of species/compounds.

Ideas for additional analyses

While somewhat outside the scope of a review, it seems to me that there are many other interesting types of analyses that could be applied to this rich data set. Here are a few suggestions:

- Contrast rates of divergence in pheromones, olfactory responses, or behavior among sympatric and allopatric species pairs...
- Examine phylogenetic signal at different depths in the tree – in addition to summarizing across whole tree as Pagel's lambda does.
- Plots of species divergence times (eg. calibrated molecular clock) versus metrics of pheromone or olfactory divergence.
- I was particularly surprised by how often the same pheromones are used by distant species. Can this be quantified? Why would two very distant species use the same male-specific compounds when few to no intervening species do? Are these compounds ancestral? Or do conserved pheromone synthesis pathways have a few possible biochemical outcomes and species simply flip back and forth through time in an 'effort' to discriminate close relatives? How often are compounds re-used? Are any totally unique? (see compounds 16, 17, 22, 38)
- More analyses focused on species groups/pairs with known interesting ecology – which would enable clearer-cut hypotheses. E.g., heightened chemical differences in sympatric species.

Other points

For the male-choice experiments, a large fraction of the tests (and significant results) involve cVA (as this male-specific compound is MUCH more widespread than any of the others). I strongly suggest separating the discussion of cVA results from that of the results for other male-specific, transferred compounds. Otherwise it is not clear how many of the conclusions are relevant just to cVA or more broadly applicable to male-specific transferred compounds in general.

This study addresses olfactory sensing, but some of these compounds are likely sensed by the taste system (as mentioned in the introduction). It would be helpful to be more clear about which compounds are known to be, or expected to be, sensed by taste pathways rather than olfactory pathways when presenting data on olfactory responses. The authors state only that most male-specific compounds are relatively small and therefore expected to be volatile. But no further details are given. Some of the compounds in the list are definitely tastants. Is there some way to note, and exclude from downstream analyses, compounds that are known tastants and/or so large that they are unlikely to be detected by antennae? I also suggest addressing how the focus on olfaction may or may not have affected conclusions in the discussion section.

Why does Supp. Fig. 1A look so much redder overall than Supp. Fig. 1A'? The legend says data were normalized to the sum of feature areas within samples, so plots should have the same mean color, even if one has more extremes. It's difficult to tell if this is the case, but the male plot certainly looks much darker... may be worth rechecking this analysis.

How is sexual mono/dimorphism in chemical profiles defined? E.g., to be called sexually monomorphic, do males and females need to have identical amounts of all compounds? Or just have identical sets of compounds (in any amount above a cutoff)?

Fig 3: I think the big plot of olfactory responses in panel b could be moved to the supplement, to make more room for the analyses in panels d-f and the illustration of the cluster analysis in the supplemental figure (which could be moved to the main figure). The analyses are more broadly interesting than the raw data. I would also like to see a version of Fig. 3b that is binary (response/no response) and combines data from the two sensilla. This would better allow the reader to evaluate at a glance how many conspecific and heterospecific compounds are detected or not (a result that is key to the authors' conclusions). From the current plot, it is impossible to tell whether undetected compounds are putative sex pheromones in a given species.

Along the same lines, the description of the analyses of olfactory responses require clarification and expansion in the text. I was confused by lines 232-243 – especially the bit about the cluster analysis and self-loops. I also couldn't find any mention of the very clear difference between at1 and at4 shown in panel 3f.

Lines 240-244: Authors note low phylogenetic signal in at1 and at4 responses and thus rapid evolution. Relative to what? Is the inference that this has to do with rapid evolution of pheromone signaling or might all sensillar responses (to both pheromones and non-pheromones) evolve at similar, 'rapid', rates?

Fig 3. How might the different numbers of neurons in at1 versus at4 sensilla alter the analytical comparisons of these two sensilla types? Can results be broken down by neuron rather than sensillum? Is it known whether the relative amplitudes of the three neurons in at3 are conserved to allow for this type of analysis? Lumping them together to generate a global picture of at3 might obscure interesting patterns.

Lines 248-253 Tests of positive selection on pheromone receptors: Some sort of description of the type of statistical test/approach used would be helpful in the main text. Looks like the authors used a branch-site model to look for elevated dN/dS at any site in the gene and along any branch. The current text makes it sound like the test was on average dN/dS across the entire gene and tree.

I can't find a description of the statistical analysis of the mating experiments aside from the mention of chi-square tests in the legend. Was a multiple test correction applied to these chi-square tests? If not, it definitely should be.

The reader is directed to Materials and Methods for further information about chromatogram analysis by XCMS and chemical identification. However, I can't seem to find this in the files provided. For example, what is meant by a "feature"? Is there a 1:1 correspondence between features and putative compounds? Also, the authors report that compound identity was confirmed by comparing mass spectra to commercial/synthesized standards. Were retention times also compared?

Minor points

In general, there are many unnecessary commas (e.g, line 225 “We, next, recorded [...]”). Sometimes they are just distracting and should be removed, but in other cases, sentences need more substantial rewriting to improve clarity (e.g., lines 62-64, 323-325).

Line 71: Is there really any evidence for neutral drift in pheromone signals?

Fig 1: Surprised to see 100% bootstrap support for branching order in the *simulans*, *mauritiana*, *sechellia* clade – which is widely considered a polytomy. This makes me wonder what high bootstrap support means for other clades involving rapid species divergence. This said, it doesn’t seem like these confidence measures have any real effect on the analyses.

Lines 150-155: Need to be careful about use of “species” and “heterospecific”. I think I know what the authors mean, but the current text is nonsensical.

Line 159: Do you mean neutral evolution or stabilizing selection?

Fig. 1b/b’, 3f: Color scale counterintuitive with blue indicating high correlation and red low. The opposite makes more sense to me.

Line 159: Why ‘a subset of male chemicals’ and not just ‘male chemicals’ or ‘at least a subset of male chemicals’. It sounds like data has already been presented implicating a specific subset of male chemicals.

Line 177: Authors say all female-specific compounds are unsaturated, but two of the structures shown are saturated.

Suppl Fig 4: What is the meaning of font color for species names (black vs grey)?

Was the solvent used for perfuming flies in the behavioral experiments DCM or hexane? The figure/methods text are inconsistent.

Reviewer #3 (Remarks to the Author):

This manuscript generates and synthesizes an extraordinary amount of data on the chemical signals produced by 99 species in the *Drosophilidae* family. Sexually dimorphic compounds that are transferred to females after copulation are examined more closely with respect to sensory receptor activation and ability to influence. In addition, the whole genome sequences for 58 species were generated as part of these studies. Undoubtedly, the data sets will be very useful not only for evolutionary biologists and chemical ecologists interested in the sexually dimorphic features of pheromone systems. The manuscript, however, suffers from an overly broad interpretation of the data. In addition, a number of

critical experimental details are missing. Finally, the manuscript needs to be edited for greater clarity.

Major critiques

Limitations of this analysis that should be addressed:

1. There are many instances in insects and mammals where a combination of molecules in precise ratios is used as a signal rather than a single compound¹. Courtship experiments in the manuscript test the activity of single, sexually dimorphic compounds and do not account for synergistic effects between combinations of pheromones. Fig 4 shows numerous instances where no behavioral change in response to conspecific perfuming is observed. It may be the case that for some of these species, multiple compounds are needed to induce a behavioral change. This is not a request for the authors to identify all bioactive compounds. However, it is important for authors to point out that this is a limitation of the study. Sex pheromones do not necessarily have to be produced in a sexually dimorphic manner and can have different modes of evolution. While the analyses reveals that a single compound can evolve rapidly, it is difficult to extrapolate this finding to sex pheromones in general, particularly ones that are not produced in a sexually dimorphic manner or are active in combination with other compounds. In particular, the section from 357-373 should be more cautiously written. The experiments in this manuscript do not exclude the possibility that non-sexually dimorphic compounds profiles also function as sex pheromones or interaction of a combination of compounds regulate mating decisions.

2. A broad assumption is made that sex pheromones are detected only by at1 and at4 sensilla, based on the van der Goes van Naters & Carlson paper. However, there is an abundance of literature on the involvement of other receptor types that mediate pheromone and olfactory communication². Therefore, the conclusions drawn about the rapid evolution of receptors compared to chemicals need to be heavily tempered by the fact that the analysis focuses only on a subset of possible pheromone receptors. Also, the conclusion that different olfactory channels are used to detect heterospecific vs conspecific signals are limited to findings pertaining to at1 and at4 and should not be generalized for other sex pheromone receptors. The section from 264-265 (“females responses does not fit the evolution of their male specific compounds”) is too broad and is not supported based on the responses of 2 sensilla types. Lines 415 – 438 should be re-written to address these limitations.

3. The female avoidance of male perfumed with cVA in species where males don't produce cVA is an interesting observation. However, it is not clear from the experiments whether females are avoiding cVA or avoiding the presence of a foreign compound. The use of a cVA analog (eg., a positional isomer) as a control would greatly strengthen these experiments.

4. Many of the taxa included in the tree are missing a lot of data, though the support numbers are robust. A higher threshold, eg. 70% data completeness, is generally used. Could authors please provide a justification as to why these taxa were included? Considering that a very small percentage of all fly species is sampled, it would be difficult to make conclusions about gains or losses. Both possibilities should be mentioned throughout the manuscript (381). Given the low data completeness, information about branch lengths should be omitted, or an arbitrary length used.

Minor corrections

I am not able to access the genomes through the NCBI Sequence Set Browser. Perhaps it may be the case that the genomes have not been uploaded yet or released for general access.

Figure 1 - The nodes should be dated so that we could see the timeframe of change/evolution of chemical signals. Branch length information should be removed.

Fig 1b shows the cross-correlation of total compound abundance between species. It is unclear what the biological significance of total abundance would be – why is this an informative feature to compare? The authors should discuss what could be the significance of a phylogenetic signal in total chemical abundance. The data shown in Sup Fig 1 seem much more compelling as a main figure.

PCA analysis: could authors please indicate which compounds were most strongly loaded for the PC1 and PC2 axes? Were these male-specific molecules?

96: This statement is overly broad - the neural processing of gustatory pheromones is not well-understood. Integration of pheromone signals with other sensory modalities is also not understood.

104: This statement is overly broad - other olfactory signals are known to play a role in sexual communication including 9-tricosene, 7-tricosene, and 7-pentacosene. While it's not clear that 7,11-nonacosadiene and 7,11-heptacosadiene are olfactory signals, these female-specific molecules are important in *D. melanogaster* sexual communication.

139 - 164: The authors state that closely related species exhibit more similar chemical profiles in males (152) but in 193, "closely related species possess dissimilar male-specific compounds". It's clearer in the discussion where authors differentiate between sexually dimorphic compounds and other cuticular compounds. However, this point should be made more explicit in this section.

154: "male species... display high correlation" – this is awkward phrasing. Chemical profiles, not males, are highly correlated.

206-208: "blend of multiple compounds" – what does this refer to? Multiple compounds that are sex specific? This should be specified.

221: sex pheromone-responsive neurons are also found in legs, maxillary palps, labellum, other sensilla in antenna

224: The following paper should be cited with respect to the response of Or47b to methyl laurate:
Lin HH, Cao DS, Sethi S, Zeng Z, Chin JSR, Chakraborty TS, Shepherd AK, Nguyen CA, Yew JY, Su CY, Wang JW. Hormonal Modulation of Pheromone Detection Enhances Male Courtship Success. *Neuron*. 2016 Jun 15;90(6):1272-1285. doi: 10.1016/j.neuron.2016.05.004. Epub 2016 Jun 2. PMID: 27263969; PMCID: PMC4911275.

The same paper shows that Or47b responds to methyl myristoleate, methyl myristate, myristoleic acid,

myristic acid, palmitoleic acid, palmitic acid.

239: what is meant by “olfactory communications”? Figure legend explains the coefficient better. In the text, it is very confusing.

246: electrical responses don’t “exhibit”

245-247: This statement can only be made for at1 and at4 sensilla but statement makes it sound like olfaction in general.

262: Assay is testing the detection of “compounds” or “chemicals”, not “males”.

262: female receptors detect male-specific compounds, not males

263: What does “not fit” mean?

329 – 332: Needs to be rewritten for clarity

335: ‘intra-sexual behavior’ should be ‘intraspecific/conspecific sexual behavior’ and ‘inter-sexual isolation’ should be ‘sexual isolation’

380: Without an ancestral state analysis, it's not always wise to just state that we're seeing multiple gains, as opposed to multiple losses, of a state. Both possibilities should be accounted for since no analysis was run.

385: the production of (...) ‘is’ and not ‘are’

492: OR appears for the first time, should be explicated

565: same for DCM (explained in figure 4 legend but not in the text)

643/644: why not test the normality of the data instead of ‘visual inspection’?

770: text for figure 3 panel (b) says that grey bubbles represent heterospecific compounds while the graphic legend in the figure gives ‘non-specific compound’ as key for the grey color. This is confusing as ‘non-specific’ could mean ‘non male-specific’

Panel (f) of the same figure is not readable, even at maximum zoom in, consider rearranging and make it bigger/better quality

794: what does it mean to “detect a species” – specify that this is looking at detection of major compounds from a given species

828-830: confusing figure key for panel (a), 'left below' or 'right below' of what? This doesn't seem to apply to this panel

834: 'left side of the horizontal dashed stoke' -> this seems to refer to the 'below' part of the panel and not the 'top' one. Also 'stoke' should be replaced with 'line'

866: XCMS – should be GCMS

884: B' refers to female profiles, not male profiles.

891: Please indicate that quantities are relative
GCMS EI spectra should be uploaded to publically available database.

Methods

A definition for what is considered to be monomorphic vs dimorphic should be provided in the methods. Does dimorphic mean the complete absence of a compound in females vs male or is it a difference in quantity between the two sexes? If the latter, provide the quantitative standard that was used when labeling a compound as dimorphic.

Please provide details about how soon chemical extracts were made after females were mated and how many flies were used for each extract.

While the diagrams in Fig 4 of the different behavioral assays is very helpful, the methods for mating assays are written in a confusing manner. Please rewrite the mating assay section so the set up is clear for competitive experiments with 2 males and 1 female; competitive experiments with 2 females and one male. Also, indicate in which instances live vs headless females are used. Lines 594-596 are particularly confusing.

Please comment on the size of the species used in the assays compared to the size of the chamber. The size of the chamber can have a drastic effect on behaviors^{3, 4}. Is a 1 cm diameter chamber sufficient for each of the different species tested?

1. Mori, K. (2007) Significance of chirality in pheromone science, *Bioorg. Med. Chem.* 15, 7505-7523.
2. Joseph, R. M., and Carlson, J. R. (2015) *Drosophila* Chemoreceptors: A Molecular Interface Between the Chemical World and the Brain, *Trends Genet* 31, 683-695.
3. Kravitz, E. A., and Fernandez, M. P. (2015) Aggression in *Drosophila*, *Behav Neurosci* 129, 549-563.
4. Griffith, L. C., and Ejima, A. (2009) Courtship learning in *Drosophila melanogaster*: diverse plasticity of a reproductive behavior, *Learning & memory* (Cold Spring Harbor, N.Y.) 16, 743-750.

Nature communications (NCOMMS-20-44553-T): Response to reviewers' comments

RESPONSE: We thank the editor and the reviewers for their careful reading of our manuscript and for their positive comments. We discuss below each of their specific concerns and how we have addressed these, where possible, with new data or analyses in the current submission to *Nature communications*. We refer in our responses to the figure and lines numbering in the new submission to aid appreciation of where we have incorporated new data and text to respond to the reviewers' concerns.

Reviewer #1 (Remarks to the Author):

The manuscript by Khallaf et al represents an impressive dataset and resource examining pheromone communications among 99 drosophilid species. The authors present massive data examining the phylogenetic relationship among 99 drosophila species, the chemical profiles of these species (as detected by GC), which of these chemicals are species specific, and which are transferred during mating. The authors also examine mating behaviors of 54 species that displayed sexually dimorphic pheromone profiles transferred during mating. Finally, the authors record from the two trichoid sensilla in 54 species to examine the extent to which chemicals are detected by the olfactory systems. Among the many findings, the authors present evidence for many new sexually dimorphic pheromones, male specific pheromones in additional species, the independent evolution of chemical profiles and the olfactory system's ability to detect these chemicals, and multiple strategies for speciation among closely related species. The data presented here provide a solid foundation for the extensive investigation of chemical communication between species and will remain a valuable resource for many years to come.

RESPONSE: Thank you very much for such a positive feedback!

I have no major concerns regarding this work, and strongly support its publication. At times the manuscript could help clarify terms that might be unfamiliar to a more general audience. Such as a description of Pagel's λ , 'low phylogenetic signal' (line 187), etc. Also, please add to the figure legend for Figure 3 when male or female at1/at3 are being recorded. The authors may wish to consider uploading their mating movies to Youtube or another service, which can be an effective method of storage and distribution.

RESPONSE: We have added now in the lines 160-164 the definition of Pagel's λ and phylogenetic signal. Moreover, we highlighted in the line 803 (figure legend for Figure 3) and in the line 626 (materials and methods section) that single sensillum recordings are carried out in females. Furthermore, we are also providing a website DOI that will include over 250GB of raw data including genome sequences (66 species), mating movies (1467 recordings for 55 *Drosophila* species; 16-48 replicates per species), chemical profiles of virgin males, and virgin and mated females (over 1500 replicates; five replicates or more of each sex in all 99 species), and other supplementary data (<https://dx.doi.org/10.17617/3.5w>).

Reviewer #2 (Remarks to the Author):

Khallaf et al examine the evolution of sex pheromones, olfactory detection pathways, and sexual behavior among up to 99 species of Drosophilid flies. *Drosophila* provides a remarkable system for addressing these phenomena, with species showing diverse ecologies and sexual signals despite relatively well known receptors and olfactory circuits. They show that even closely related species often differ in sexual signaling and behavior and test for levels of phylogenetic constraint in various features of the system.

The work that went into this study, and the richness of the data produced, are truly impressive – chemical analysis of 5 males and 5 females from each of 99 species! Extracellular recordings from female antennal sensilla in ~50 species! Mating behavioral assays in ~50 species! These data will provide a huge resource for the *Drosophila* community.

The analyses performed are less strong, in my opinion, than the data themselves. The questions and hypotheses are often ambiguous, making it difficult to compare specific results to expected outcomes and interpret accordingly. As a result, interpretations sometimes felt arbitrary, surface-level, and/or poorly supported by the data.

Overall, I think the data are important and the work is potentially publishable, but I recommend a significant revision, involving (1) clarification of hypotheses and expected outcomes and (2) more careful and measured interpretations. I would also like to see (3) new, more incisive analyses that are a better match for this amazing dataset and the newly clarified hypotheses/predictions.

RESPONSE: We thank the reviewer for these positive and very constructive comments. We have thoroughly revised our manuscript according to all suggestions and have tried to clarify all points by addressing them, where possible, with new data or analyses.

Technical issues

I think the study is technically sound. My only substantial comment is that vapor-phase concentrations of the stimuli used for the single sensillum recordings are not clear. Were vapor-phase concentrations of delivered stimuli directly measured to ensure biological relevance? If not, how do the authors know that the concentrations are appropriate? And how does uncertainty in this regard affect conclusions? It seems to me that the most important issue is whether some fraction of biologically relevant olfactory responses may have been missed due to unnaturally low vapor-phase concentrations (e.g. for compounds with limited volatility). The various circle sizes in Figure 3b also give the impression that quantitative differences in the response of a given sensillum to different compound reflect neural sensitivity, when they may instead reflect differences in compound volatility (and consequent variation in vapor-phase concentration).

Response: This is an excellent point and we agree that olfactory responses to a given compound could also be influenced by its vapor pressure. Therefore, we searched the web for the vapor pressures of the 36 compounds that we used for the single sensillum recordings and found that vapor pressure of 29 compounds at 25 °C are available. We then correlate these measures to the number of the olfactory responses that compounds elicit in at1 and at4 sensilla of the 54 species. Surprisingly, the compounds' vapor pressures have no impact on the olfactory responses (**new Supplementary Figure 3D**). These new results support our earlier findings (**Fig. 3b**) that these male-specific compounds elicit neuronal responses due to their high affinity to the olfactory receptors, but not due to the compounds' high volatility (information added to **Supplementary Table 4**).

Our preliminary electrophysiological recordings and the dose-dependent responses of at1 and at4 neurons in a previous study (Khallaf et al., 2020) revealed that high concentrations

of odorants (e.g., 10^{-1} dilution (v/v)) elicit strong responses that might saturate or kill the olfactory neurons, while low concentration of odorants (e.g., 10^{-5} dilution (v/v)) elicit no or low responses. Moreover, due to high number of tested compounds and species, dose-dependent responses for the different compounds could not be performed. Therefore, an intermediate concentration (10^{-3}) has been used for all odorants used in the single sensillum recordings. We have added this information to the Materials and Methods section and figure legend (lines 642- 646).

Supplementary Figure 3D. A log-log plot of compounds' vapor pressure at 25 °C and the number of their olfactory responses in at1 and at4 of the 54 *Drosophila* species, indicating that both variables are uncorrelated.

Conceptual issues

My most significant critique of the study has to do with the conceptual foundation and interpretations. The authors frame the study as a test of rapid coevolution between sex pheromones and their cognate sensory channels, but it's not clear to me exactly which analysis tests for coevolution. We learn about the degree to which various aspects of species and sex specific signals are phylogenetically constrained. We gain a ton of valuable information about what pheromones are produced, and what neurons may be used to detect them, and how they may affect mating behavior in a staggering number of species. But it is not totally clear what the key hypotheses are (eg. coevolution vs no coevolution? rapid vs slow evolution?...). The analyses are largely descriptive – rather than testing specific, well-defined hypotheses and predicted outcomes. If the hypotheses pertain to evolutionary rates and phylogenetic constraint, then what serves as the reference point (i.e. neutral expectation) for saying that something is evolving rapidly? Or that something is coevolving?

Response: Following the reviewer's suggestion, we have now highlighted the hypothesis and the key question of this section (lines 151-152 and lines 164-165). Furthermore, we realized that we do not have a good model to test whether the chemical traits are positively selected

or neutrally evolved. Therefore, we tuned down, and in some cases deleted, many of our propositions about the rapid evolutionary rates.

The first analysis of chemical signals comes closer to testing specific hypotheses than other analyses in the paper. The authors conduct an interesting comparison between the rates of evolution of male vs. female compounds, and, within male compounds, of male-specific versus shared compounds. However, the interpretations are internally inconsistent. When comparing male signals to female signals, the reduced level of phylogenetic constraint for female compounds (Supp fig 1d) is interpreted as showing that they are less important for species-specific behaviors. However, when comparing male-specific compounds to other compounds found on males, the reduced level of phylogenetic constraint for male-specific compounds is interpreted as rapid evolution via positive selection and saltational shifts. Why the different interpretations of the same type of comparative signal? I recommend conducting a single, more wholistic analysis of three types of compounds... female-specific, male-specific, and shared. The distribution of Pagel's lambda for these three could be shown together in the same plot and discussed at the same time. Alternative explanations should be considered. For example, decreased Pagel's lambda (reduced phylogenetic constraint) could result from both increased positive selection or decreased stabilizing selection. The authors should also consider other analyses that may allow them to distinguish among these hypotheses (see some suggestions below).

Response: We agree that the male-specific chemicals do not differ in lambda distribution compared to the other male compounds, and both of them are more right-skewed than female compounds. We, therefore, deleted the inconsistent sentences and rewrote other sentences (e.g., lines 160-164; lines 194-200; lines 239-243; lines 350-353; lines 376-379).

Another place where the data are extremely rich, but it is not clear to me what the prediction is and, whether it has been upheld, is in the discussion/analysis of detection of pheromones by at1 and at4 sensilla. Fig. 3b represents an impressive amount of work. So much information is contained there. The authors note that females of more than 75% of species can detect compounds on conspecific males through these sensilla – suggesting positive selection. Where is the test of positive selection here? Is it Pagel's lambda? What value of Pagel's lambda do we expect to see if there is no positive selection? They then go on to show that females of many species also detect compounds found on heterospecific males through these same sensilla – discussing the idea of evolutionary labile central circuits toggling the behavioral valence of conserved peripheral signals. A single species is given as an example, and a very interesting anecdote, but I can't find a broad test of this hypothesis anywhere in the analysis. For example, when closely related species diverge in male-compounds, how often is this accompanied by a change in the female sensillum that detects those compounds... and/or by a shift in behavioral valence? Is the ability of females to detect heterospecific compounds largely driven by retention of detection ability for compounds that are produced by other species in the same clade? Or is detection ability more widely dispersed across *Drosophila* phylogeny – and, perhaps, better explained by sympatry than phylogeny?

Response: We realize that these propositions may have come across more strongly than is merited by the data; we thank the reviewer for raising this issue, which – together with new data, described below – have encouraged us to re-think through this argument. First, we agree that the ability of females of more than 75% of species to detect their male-specific compounds does not support that conclusion of the positive selection on the neuronal level. However, by assessing the selection pressures on the pheromone receptors, we revealed an evidence of positive selection on all these genes with the highest pressures on the *Or47b* and the *Or67d* loci. In the light of the positive selection on the receptor level, we deleted the old lines 249-250 and added the lines 264-266.

Second, following the reviewer's suggestion, we added new data (new Supplementary Table 6 (sheet2)), where we tested how often divergence in male-specific

compounds is accompanied by a change in the neuronal responses of the females. Both an overall phylogenetic generalized least squares (PGLS) estimates – that explain the phylogenetic corrected correlations between the production level of male-specific compounds and their neuronal response in at1 and at4, and a test using 16 pairs of closely related species – revealed almost no correlations between the evolutionary changes in male chemicals and female responses in these sister species. This indicates that the differential females' olfactory responses are not correlated with the variations in the male chemical traits. This finding is paradigm-shifting because now we know that reproductive isolation can be in part driven by heterospecific discrimination rather than conspecific preference. We have added these findings in the results and discussion section (lines 270-274 and 439-443).

Finally, I was confused about the hypothesis being tested by the female-choice mating assays described in Fig 4b (and lines 284-300). What is achieved by adding more of a male-specific compound to a male that already has this compound? And then testing him against an unmanipulated male that also presumably has this compound? Is this a test of whether quantitative levels of male-specific compounds affect mating? And what is the prediction? Should we expect elevated levels of these compounds to be more attractive to females than their natural levels? It is not clear what these experiments are telling us about fly biology. Unless the authors can significantly clarify the predictions/expected outcomes/interpretations, I suggest removing these experiments from the study and focusing on the more-interpretable male-choice experiments (Fig. 4c). However, because SSR was performed only on females (as far as I can tell), the authors may need to find another way to justify the choice of species/compounds.

Response: Previous reports have shown that the amount of the cuticular hydrocarbons increases by age (Revadi et al., 2015; Snellings et al., 2018). Moreover, older males possess higher amounts of male-specific compounds – such as cVA and 7-tricosene in *D. melanogaster* and *D. sukijii*, respectively (Bartelt et al., 1985; Snellings et al., 2018), and (Z)-10-heptadecen-2-yl acetate and (Z,Z)-19,22-octacosadien-1-yl acetate in *D. mojavensis* (Khallaf et al., 2020). Furthermore, *Drosophila* females exhibit a preference to copulate with older males (Dweck et al., 2015; Khallaf et al., 2020; Lin et al., 2016). Therefore, we hypothesized that males perfumed with single male-specific compounds would have higher copulation advantage than the solvent perfumed males (which carried the same compound but a lower amount of it). Our hypothesis turned to be true in 11 instances, where females preferred to copulate with the male-specific compound perfumed males compared to solvent control males. While, it had an opposite effect in 5 instances and no impact in 29 other instances. We believe that these results are important, as they reveal that females of *Drosophila* species interpret the increase of the pheromones cues differently resulting in an opposite impact on the female sexual receptivity. However, we agree with the reviewer's comment that the hypothesis of these experiments was unclear, therefore, we rewrote the lines 300-304 and explicitly highlighted our purpose of these experiments.

Ideas for additional analyses

While somewhat outside the scope of a review, it seems to me that there are many other interesting types of analyses that could be applied to this rich data set. Here are a few suggestions:

RESPONSE: We thank the reviewer for suggesting these thoughtful ideas, which – together with the new data, have encouraged us to conduct comprehensive analyses, as follow:

- Contrast rates of divergence in pheromones, olfactory responses, or behavior among sympatric and allopatric species pairs...

RESPONSE: We tested contrast rates of divergence in male-specific compounds, and how this is accompanied by a change in the neuronal responses of the females. We added this new dataset to the manuscript as a **new Supplementary Table 6 (sheet2)**.

- Examine phylogenetic signal at different depths in the tree – in addition to summarizing across whole tree as Pagel’s lambda does.

RESPONSE: Following the reviewer’s suggestion, we measured the phylogenetic signal of each male and female compounds at different depth by splitting the phylogeny (**Fig. 1A**) into 9 different trees to cover the different *Drosophila* groups (repleta, virilis, melanica, cardini, immigrans, melanogaster, obscura, willistoni, and saltans), which are highlighted in different colors in **Fig. 1A**). We provide this huge dataset as a **new Supplementary Table 2**, which contains of additional 9 sheets (one sheet for male and female compounds in each group), and 10 nexus files, which represent the 99sp phylogeny and 9 subtrees, as **new Supplementary Files 1-9**. We thank the reviewer for her/his suggestion, as we believe that this new dataset will aid specialists who are interested to look into the phylogenetic signal of male and female compounds in the different groups.

- Plots of species divergence times (eg. calibrated molecular clock) versus metrics of pheromone or olfactory divergence.

RESPONSE: We provide a new calibrated tree dated according to the previously published calibration points in (Obbard et al., 2012; Russo et al., 2013) as **new Supplementary Figure 1D** (see below).

Supplementary Figure 1D: Dated phylogeny of 99 species within the family Drosophilidae inferred from 13,433,544 amino acids sites that represent 11,479 genes.

- I was particularly surprised by how often the same pheromones are used by distant species. Can this be quantified? Why would two very distant species use the same male-specific compounds when few to no intervening species do? Are these compounds ancestral? Or do conserved pheromone synthesis pathways have a few possible biochemical outcomes and species simply flip back and forth through time in an 'effort' to discriminate close relatives? How often are compounds re-used? Are any totally unique? (see compounds 16, 17, 22, 38)

RESPONSE: We are certainly also very surprised by the fact that many of the male-specific compounds exhibit low phylogenetic signals (**old Fig. 2d; now new Supplementary Figure 2C**), and thus are repeatedly present across distant species, while few are exclusively species- or group-specific compounds (**Fig. 2b; Supplementary Figure 2B**). Probably, some of the distant species might possess conserved pheromonal biosynthetic pathways, however, testing this proposition is outside of our expertise and is out of the scope of this current study. To make this section clearer to the reader, we highlighted and re-wrote the **lines 194-199**.

- More analyses focused on species groups/pairs with known interesting ecology – which would enable clearer-cut hypotheses. E.g., heightened chemical differences in sympatric species.

RESPONSE: We agree with the reviewer that a better understanding of the ecological niches and the geographical distribution of these *Drosophila* species would advance our understanding about the driving forces, which shape the evolution of sex pheromone communication systems. However, as it is not easy to find enough ecological information for many of those species and as we feel that these additional analyses would make the manuscript even more complex, we hope that future ecological investigations will be conducted to explore these issues in greater detail.

Other points

For the male-choice experiments, a large fraction of the tests (and significant results) involve cVA (as this male-specific compound is MUCH more widespread than any of the others). I strongly suggest separating the discussion of cVA results from that of the results for other male-specific, transferred compounds. Otherwise it is not clear how many of the conclusions are relevant just to cVA or more broadly applicable to male-specific transferred compounds in general.

RESPONSE: We appreciate the reviewer's suggestion. We have now separated the roles of cVA in the male-choice experiments.

This study addresses olfactory sensing, but some of these compounds are likely sensed by the taste system (as mentioned in the introduction). It would be helpful to be more clear about which compounds are known to be, or expected to be, sensed by taste pathways rather than olfactory pathways when presenting data on olfactory responses. The authors state only that most male-specific compounds are relatively small and therefore expected to be volatile. But no further details are given. Some of the compounds in the list are definitely tastants. Is there some way to note, and exclude from downstream analyses, compounds that are known tastants and/or so large that they are unlikely to be detected by antennae? I also suggest addressing how the focus on olfaction may or may not have affected conclusions in the discussion section.

RESPONSE: We have now addressed the reviewer's suggestion – not only female-specific compounds but also some of the male- compounds could be detected by gustatory receptors. We also agree that the conclusion that different chemosensory channels are used to detect heterospecific vs conspecific signals are limited to findings pertaining to at1 and at4. Therefore, we addressed this limitation by re-writing the **lines 270-274**.

Why does Supp. Fig. 1A look so much redder overall than Supp. Fig. 1A'? The legend says data were normalized to the sum of feature areas within samples, so plots should have the same mean color, even if one has more extremes. It's difficult to tell if this is the case, but the male plot certainly looks much darker... may be worth rechecking this analysis.

RESPONSE: The color impression is determined by some extreme values in both maps and unfortunately, we cannot define the scale in the program. Therefore, we added a note to the caption that the scale of both figures is different.

How is sexual mono/dimorphism in chemical profiles defined? E.g., to be called sexually monomorphic, do males and females need to have identical amounts of all compounds? Or just have identical sets of compounds (in any amount above a cutoff)?

RESPONSE: When both sexes in a given species share similar chemical compounds regardless the compounds' quantities, this species is identified as a monomorphic species. We have now added more info about the definition in the Materials and Methods section (lines 176-180).

Fig 3: I think the big plot of olfactory responses in panel b could be moved to the supplement, to make more room for the analyses in panels d-f and the illustration of the cluster analysis in the supplemental figure (which could be moved to the main figure). The analyses are more broadly interesting than the raw data. I would also like to see a version of Fig. 3b that is binary (response/no response) and combines data from the two sensilla. This would better allow the reader to evaluate at a glance how many conspecific and heterospecific compounds are detected or not (a result that is key to the authors' conclusions). From the current plot, it is impossible to tell whether undetected compounds are putative sex pheromones in a given species.

RESPONSE: Following the reviewer's suggestion, we have generated a new figure, where we combined the neurophysiological data of at1 and at4 responses (Reviewer figure 1). However, we still believe that the information, whether or not a compound is detected by at1 or at4 is important and, hence, would like to keep the figure as it is.

Reviewer Fig. 1: Color-coded electrophysiological responses towards heterospecific compounds (grey bubbles) and conspecific compounds (colored bubbles) in at1 (top) and at4 (bottom) sensilla of females of 54 species.

Along the descriptor

olfactory responses require clarification and expansion in the text. I was 232-243 – especially the bit about the cluster analysis and self-loops. I also couldn't find any mention of the very clear difference between at1 and at4 shown in panel 3f.

RESPONSE: We have now added more information in the results section about olfactory responses (lines 249-251). We also moved Fig. 3f to make it bigger with higher quality to new Figure supplementary 3C.

Lines 240-244: Authors note low phylogenetic signal in at1 and at4 responses and thus rapid evolution. Relative to what? Is the inference that this has to do with rapid evolution of pheromone signaling or might all sensillar responses (to both pheromones and non-pheromones) evolve at similar, 'rapid', rates?

RESPONSE: Indeed, to draw such conclusion, we need to compare the evolutionary rates of pheromone and non-pheromone (i.e., food) receptors. Therefore, we deleted this sentence ~~“, indicating that olfactory channels of the different species evolve rapidly apart from their phylogenetic relationships”~~.

Fig 3. How might the different numbers of neurons in at1 versus at4 sensilla alter the analytical comparisons of these two sensilla types? Can results be broken down by neuron rather than sensillum? Is it known whether the relative amplitudes of the three neurons in at3 are conserved to allow for this type of analysis? Lumping them together to generate a global picture of at3 might obscure interesting patterns.

RESPONSE: We would love to be able to sort the neurons in the trichoid sensilla, so that we can assign the olfactory responses to a specific neuron. However, this has proven to be technically very challenging due to two reasons: **1)** number of neurons per same sensillum type is not conserved in the different *Drosophila* species – as revealed by number of at1 neurons across the different species in Supplementary Figure 3A. **2)** Apart from *D. melanogaster*, which is known to be a neurophysiological model organism and its olfactory sensory neurons exhibit differential spike amplitude, it is very hard to distinguish the number of neurons, specifically in at4 sensillum, based only on the amplitude. We have now addressed this limitation in lines 653-657 ~~“Note that number of neurons per same sensillum type is not conserved in the different *Drosophila* species – as revealed by number of at1 neurons across the different species in Supplementary Figure 3A. Moreover, sorting the number of neurons based on the spike amplitudes in all at4 and some at1 sensilla is technically challenging due to the close spike amplitudes of the sensillum neurons.”~~.

Lines 248-253 Tests of positive selection on pheromone receptors: Some sort of description of the type of statistical test/approach used would be helpful in the main text. Looks like the authors used a branch-site model to look for elevated dN/dS at any site in the gene and along any branch. The current text makes it sound like the test was on average dN/dS across the entire gene and tree.

RESPONSE: We have now added a description of the statistical test in line 261.

I can't find a description of the statistical analysis of the mating experiments aside from the mention of chi-square tests in the legend. Was a multiple test correction applied to these chi-square tests? If not, it definitely should be.

RESPONSE: We indeed used only the chi-square test. As we did not compare the effect of different compounds but only tested, whether a male with one specific compound added becomes more attractive than the control male, we still believe that no correction for repeated measurements is needed.

The reader is directed to Materials and Methods for further information about chromatogram analysis by XCMS and chemical identification. However, I can't seem to find this in the files provided. For example, what is meant by a “feature”? Is there a 1:1 correspondence between features and putative compounds? Also, the authors report that compound identity was

confirmed by comparing mass spectra to commercial/synthesized standards. Were retention times also compared?

RESPONSE: We apologize for this and have now added more information about the XCMS and chemical identification to the figure legend lines 884-885 and the materials and methods section lines 575-577. We have also added to line 565 that retention times of analytical standards were compared to the identified compounds.

Minor points

In general, there are many unnecessary commas (e.g, line 225 “We, next, recorded [...]”). Sometimes they are just distracting and should be removed, but in other cases, sentences need more substantial rewriting to improve clarity (e.g., lines 62-64, 323-325).

RESPONSE: We apologize for this and have now revised the mentioned sentences.

Line 71: Is there really any evidence for neutral drift in pheromone signals?

RESPONSE: We have changed the sentence to “Diversification of sex pheromones among species arises either via sexual, and/or natural selection”.

Fig 1: Surprised to see 100% bootstrap support for branching order in the *simulans*, *mauritiana*, *sechellia* clade – which is widely considered a polytomy. This makes me wonder what high bootstrap support means for other clades involving rapid species divergence. This said, it doesn’t seem like these confidence measures have any real effect on the analyses.

RESPONSE: We agree with the reviewer’s comment that branching node of *D. simulans*, *D. mauritiana*, and *D. sechellia* is considered as polytomy and our data supports this hypothesis, as shown by the very short branch length (almost zero). However, due to the huge amount of data small bias in the random sampling of sites will make the bootstrap value very high. We agree that the short branch would not largely affect subsequent analyses.

Lines 150-155: Need to be careful about use of “species” and “heterospecific”. I think I know what the authors mean, but the current text is nonsensical.

RESPONSE: Thanks for pointing this. We have now changed “heterospecific groups” to “chemical profiles of males of different groups”.

Line 159: Do you mean neutral evolution or stabilizing selection?

RESPONSE: We have deleted this sentence and changed “neutral evolution” to “stabilizing selection” in the other sentences.

Fig. 1b/b’, 3f: Color scale counterintuitive with blue indicating high correlation and red low. The opposite makes more sense to me.

RESPONSE: We have tried different color regimes, but we would like to keep it as it is.

Line 159: Why ‘a subset of male chemicals’ and not just ‘male chemicals’ or ‘at least a subset of male chemicals’. It sounds like data has already been presented implicating a specific subset of male chemicals.

RESPONSE: We have deleted this sentence.

Line 177: Authors say all female-specific compounds are unsaturated, but two of the structures shown are saturated.

RESPONSE: We have corrected this sentence.

Suppl Fig 4: What is the meaning of font color for species names (black vs grey)?

RESPONSE: We have added a description for the font color in the corresponding figure legend lines 982-983 "Species in black are dimorphic species, while in grey are monomorphic species"

Was the solvent used for perfuming flies in the behavioral experiments DCM or hexane? The figure/methods text are inconsistent.

RESPONSE: We apologize for this and now have corrected this inconsistency.

Reviewer #3 (Remarks to the Author):

This manuscript generates and synthesizes an extraordinary amount of data on the chemical signals produced by 99 species in the Drosophilidae family. Sexually dimorphic compounds that are transferred to females after copulation are examined more closely with respect to sensory receptor activation and ability to influence. In addition, the whole genome sequences for 58 species were generated as part of these studies. Undoubtedly, the data sets will be very useful not only for evolutionary biologists and chemical ecologists interested in the sexually dimorphic features of pheromone systems. The manuscript, however, suffers from an overly broad interpretation of the data. In addition, a number of critical experimental details are missing. Finally, the manuscript needs to be edited for greater clarity.

RESPONSE: We thank the reviewer for these extremely positive comments. We have thoroughly revised and rewrote our manuscript according to all suggestions and have tried to clarify all points by addressing them, where possible, with new data.

Major critiques

Limitations of this analysis that should be addressed:

1. There are many instances in insects and mammals where a combination of molecules in precise ratios is used as a signal rather than a single compound¹. Courtship experiments in the manuscript test the activity of single, sexually dimorphic compounds and do not account for synergistic effects between combinations of pheromones. Fig 4 shows numerous instances where no behavioral change in response to conspecific perfuming is observed. It may be the case that for some of these species, multiple compounds are needed to induce a behavioral change. This is not a request for the authors to identify all bioactive compounds. However, it is important for authors to point out that this is a limitation of the study. Sex pheromones do not necessarily have to be produced in a sexually dimorphic manner and can have different modes of evolution. While the analyses reveals that a single compound can evolve rapidly, it is difficult to extrapolate this finding to sex pheromones in general, particularly ones that are not produced in a sexually dimorphic manner or are active in combination with other compounds. In particular, the section from 357-373 should be more cautiously written. The experiments in this manuscript do not exclude the possibility that non-sexually dimorphic compounds profiles also function as sex pheromones or interaction of a combination of compounds regulate mating decisions.

RESPONSE: We totally agree with the reviewer criticism. Therefore, we rewrote the mentioned sections in the light of the reviewer's suggestions (lines 332-335).

2. A broad assumption is made that sex pheromones are detected only by at1 and at4 sensilla, based on the van der Goes van Naters & Carlson paper. However, there is an abundance of literature on the involvement of other receptor types that mediate pheromone and olfactory communication². Therefore, the conclusions drawn about the rapid evolution of receptors compared to chemicals need to be heavily tempered by the fact that the analysis focuses only on a subset of possible pheromone receptors. Also, the conclusion that different olfactory channels are used to detect heterospecific vs conspecific signals are limited to findings pertaining to at1 and at4 and should not be generalized for other sex pheromone receptors. The section from 264-265 ("females responses does not fit the evolution of their male specific compounds") is too broad and is not supported based on the responses of 2 sensilla types. Lines 415 – 438 should be re-written to address these limitations.

RESPONSE: We thank the reviewer for her/his very constructive comment. We have now addressed these limitations (lines 270-274) and rewrote the whole section (lines 440-444).

3. The female avoidance of male perfumed with cVA in species where males don't produce cVA is an interesting observation. However, it is not clear from the experiments whether females are avoiding cVA or avoiding the presence of a foreign compound. The use of a cVA analog (eg., a positional isomer) as a control would greatly strengthen these experiments.

RESPONSE: We realize that these propositions may have come across more strongly than is merited by the data; we thank the reviewer for raising this issue, which have encouraged us to re-think through this argument. Following the reviewer request, we provide new data (new Supplementary Figure 4C; see below), where females of the 8 species – which exhibit avoiding to mate with cVA perfumed males – have the chance to choose between two males, one perfumed with a cVA-like compound that activates the same sensillum type that cVA activates and another control male (perfumed with solvent). Similarly, females of 7 species avoided to copulate with the cVA-like compound. However, only females of *D. sturtevantii* had no preference between the rival males. We suggest that (*E*)- β -Farnesene activates extra (or different) olfactory channels than the cVA-sensing channels, the matter that may explain this discrepancy in the behavioral outputs.

Supplementary Figure 4C. Top: Schematic of a mating arena where a female of each species had the choice to mate with two conspecific males perfumed with olfactory-detected heterospecific compound or solvent (DCM). Note that we only tested the species that do not produce but still detect the heterospecific compound. Below: bar plots represent the percentages of copulation success of the rival males. Results from females that were only courted by one male were excluded.

4. Many of the taxa included in the tree are missing a lot of data, though the support numbers are robust. A higher threshold, eg. 70% data completeness, is generally used. Could authors please provide a justification as to why these taxa were included? Considering that a very small percentage of all fly species is sampled, it would be difficult to make conclusions about gains or losses. Both possibilities should be mentioned throughout the manuscript (381). Given the low data completeness, information about branch lengths should be omitted, or an arbitrary length used.

RESPONSE: The level of missing data in certain taxa is due to a relatively large variation in the pooled proportions of the sequencing libraries and the lower complexity of certain libraries. Setting the missing threshold too high would result in discarding too many alignment columns and useful data, so we chose to retain as much data as possible. Previous simulation studies have shown that Maximum likelihood methods work well when even some taxa have more than 90% of data missing, as long as the absolute amount of data has sufficient overlap with other taxa. Our tree is still strongly supported despite such variations in coverage. According to the availability in the Drosophila stock center, the chosen species were selected to cover most of (sub-) groups of the Drosophila genus (line 126). We have also replaced “gaining” by “modulation” in line 400.

Minor corrections

I am not able to access the genomes through the NCBI Sequence Set Browser. Perhaps it may be the case that the genomes have not been uploaded yet or released for general access.

RESPONSE: The bioproject link to the data is <https://www.ncbi.nlm.nih.gov/bioproject/669609>. We have instructed NCBI to release the data. However, the link may appear unavailable since it depends on NCBI to release data which we have no control over. We can request NCBI to make the data available on the date of print.

Figure 1 - The nodes should be dated so that we could see the timeframe of change/evolution of chemical signals. Branch length information should be removed.

RESPONSE: We provide a new calibrated tree dated according to the previously published calibration points in (Obbard et al., 2012; Russo et al., 2013) as new Supplementary Figure 1D.

Supplementary Figure 1D: Dated phylogeny of 99 species within the family Drosophilidae inferred from 13,433,544 amino acids sites that represent 11,479 genes. Estimated divergence times are from (Obbard et al., 2012; Russo et al., 2013).

Fig 1D shows the cross-correlation of total compound abundance between species. It is unclear what the biological significance of total abundance would be – why is this an informative feature to compare? The authors should discuss what could be the significance of a

phylogenetic signal in total chemical abundance. The data shown in Sup Fig 1 seem much more compelling as a main figure.

RESPONSE: The overall peak areas of the 248 male and 265 female chemical features, which were detected across the 99 species, were used to increase the richness and dimensionality of the pairwise correlation analyses and to test whether the males and females of the species possess similar chemical compounds in respect to quality and quantity.

Following the reviewer's suggestion, we moved the **Supplementary Figure 1B** and **1B'** to the main figure to be **Fig. 1c** and **1c'**.

PCA analysis: could authors please indicate which compounds were most strongly loaded for the PC1 and PC2 axes? Were these male-specific molecules?

RESPONSE: We have checked the variables that are strongly loaded in PC1 and PC2 and found high loadings, which is attributed due to the richness of the data (e.g., compounds that are strongly loaded in the PC1 of the female chemical profiles (**Reviewer figure 2**)). Therefore, we believe that this info could be redundant and will distract the reader.

Reviewer Fig. 2. Loadings of the female chemical compounds on PC1.

96: This statement is overly broad - the neural processing of gustatory pheromones is not well-understood. Integration of pheromone signals with other sensory modalities is also not understood.

RESPONSE: We have edited the sentence and replaced “well” by “largely”.

104: This statement is overly broad - other olfactory signals are known to play a role in sexual communication including 9-tricosene, 7-tricosene, and 7-pentacosene. While it's not clear that 7,11-nonacosadiene and 7,11-heptacosadiene are olfactory signals, these female-specific molecules are important in *D. melanogaster* sexual communication.

RESPONSE: We have edited and deleted some parts of the sentence to be “Olfactory sexual communication in *D. melanogaster* is arguably one of the best studied systems in animals²⁷, and is carried out through limited chemical signals, including cis-vaccenyl acetate (cVA)²⁸.”

139 - 164: The authors state that closely related species exhibit more similar chemical profiles in males (152) but in 193, “closely related species possess dissimilar male-specific compounds”. It's clearer in the discussion where authors differentiate between sexually dimorphic compounds and other cuticular compounds. However, this point should be made more explicit in this section.

RESPONSE: We have now edited the section and made this point clearer (lines 176-180).

154: “male species... display high correlation” – this is awkward phrasing. Chemical profiles, not males, are highly correlated.

RESPONSE: We have changed the sentence according to the reviewer suggestion.

206-208: “blend of multiple compounds” – what does this refer to? Multiple compounds that are sex specific? This should be specified.

RESPONSE: We changed it to “multiple male-specific compounds”.

221: sex pheromone-responsive neurons are also found in legs, maxillary palps, labellum, other sensilla in antenna

RESPONSE: We have changed the sentence to be “olfactory sex pheromone-responsive neurons”.

224: The following paper should be cited with respect to the response of Or47b to methyl laurate:

Lin HH, Cao DS, Sethi S, Zeng Z, Chin JSR, Chakraborty TS, Shepherd AK, Nguyen CA, Yew JY, Su CY, Wang JW. Hormonal Modulation of Pheromone Detection Enhances Male Courtship Success. *Neuron*. 2016 Jun 15;90(6):1272-1285. doi: 10.1016/j.neuron.2016.05.004. Epub 2016 Jun 2. PMID: 27263969; PMCID: PMC4911275. The same paper shows that Or47b responds to methyl myristoleate, methyl myristate, myristoleic acid, myristic acid, palmitoleic acid, palmitic acid.

RESPONSE: We apologize for the oversight and now added the reference.

239: what is meant by “olfactory communications”? Figure legend explains the coefficient better. In the text, it is very confusing.

RESPONSE: We have replaced “olfactory communication” by “olfactory interactions”

246: electrical responses don’t “exhibit”

RESPONSE: We changed the sentence to “Pairwise correlation and statistical analyses revealed that electrophysiological responses at1 and at4 neurons of the different species have low phylogenetic signals”.

245-247: This statement can only be made for at1 and at4 sensilla but statement makes it sound like olfaction in general.

RESPONSE: We have specified the sentence to at1 and at4.

262: Assay is testing the detection of “compounds” or “chemicals”, not “males”.

RESPONSE: We have added “compounds” to the sentence.

262: female receptors detect male-specific compounds, not males

RESPONSE: Here, we disagree with the reviewer and would like to keep the sentence as it is “females of 36 out of 49 species detect their male-specific compounds – (Fig. 3c),”

263: What does “not fit” mean?

RESPONSE: We have changed it to “does not correlate with”

329 – 332: Needs to be rewritten for clarity

RESPONSE: We rewrote the sentence to be “Females of *Drosophila* species, which are able to detect cVA as heterospecific signal (i.e., their males do not produce it), had the choice to mate with two conspecific males perfumed with cVA or solvent.”

335: ‘intra-sexual behavior’ should be ‘intraspecific/conspecific sexual behavior’ and ‘inter-sexual isolation’ should be ‘sexual isolation’

RESPONSE: We have changed the words according to the reviewer’s suggestion.

380: Without an ancestral state analysis, it’s not always wise to just state that we’re seeing multiple gains, as opposed to multiple losses, of a state. Both possibilities should be accounted for since no analysis was run.

RESPONSE: We have edited the sentence.

385: the production of (...) ‘is’ and not ‘are’

RESPONSE: We apologize for this mistake.

492: OR appears for the first time, should be explicated

RESPONSE: We have introduced this term in **line 105**.

565: same for DCM (explained in figure 4 legend but not in the text)

RESPONSE: We have now written it in the text.

643/644: why not test the normality of the data instead of ‘visual inspection’?

RESPONSE: We have followed the reviewer’s suggestion and have now performed a Shapiro-Wilks test for normality and found no change in our conclusions (**lines 673-674**).

770: text for figure 3 panel (b) says that grey bubbles represent heterospecific compounds while the graphic legend in the figure gives ‘non-specific compound’ as key for the grey color. This is confusing as ‘non-specific’ could mean ‘non male-specific’ Panel (f) of the same figure is not readable, even at maximum zoom in, consider rearranging and make it bigger/better quality

RESPONSE: We have changed it to “Heterospecific compounds”.

Following the reviewer’s request, we have moved **Fig. 3f** to **Supplementary Figure 3C**.

794: what does it mean to “detect a species” – specify that this is looking at detection of major compounds from a given species

RESPONSE: To clarify the meaning, we changed it to “species-specific compounds”

828-830: confusing figure key for panel (a), ‘left below’ or ‘right below’ of what? This doesn’t seem to apply to this panel

RESPONSE: We apologize for this confusion.

834: 'left side of the horizontal dashed stroke' -> this seems to refer to the 'below' part of the panel and not the 'top' one. Also 'stroke' should be replaced with 'line'

RESPONSE: We corrected these mistakes.

866: XCMS – should be GCMS

RESPONSE: XCMS is a bioinformatics software designed for statistical analysis of mass spectrometry data (Huan et al., 2017); we have now added its description to the figure legend lines 884-885 and the materials and methods section lines 575-577.

884: B' refers to female profiles, not male profiles.

RESPONSE: Thanks for this note.

891: Please indicate that quantities are relative GCMS EI spectra should be uploaded to publically available database.

RESPONSE: We added this info. We are also providing a website DOI that will include over 250GB of raw data including genome sequences (66 species), mating movies (1467 recordings for 55 *Drosophila* species; 16-48 replicates per species), chemical profiles of virgin males, and virgin and mated females (over 1500 replicates; five replicates or more of each sex in all 99 species), and other supplementary data (<https://dx.doi.org/10.17617/3.5w>).

Methods

A definition for what is considered to be monomorphic vs dimorphic should be provided in the methods. Does dimorphic mean the complete absence of a compound in females vs male or is it a difference in quantity between the two sexes? If the latter, provide the quantitative standard that was used when labeling a compound as dimorphic.

RESPONSE: Dimorphic is identified as a complete absence of the male- or female-specific compound in the other sex. We have now added this information in the results section lines 176-180.

Please provide details about how soon chemical extracts were made after females were mated and how many flies were used for each extract.

RESPONSE: We have now added this information in the material and methods section line 552.

While the diagrams in Fig 4 of the different behavioral assays is very helpful, the methods for mating assays are written in a confusing manner. Please rewrite the mating assay section so the set up is clear for competitive experiments with 2 males and 1 female; competitive experiments with 2 females and one male. Also, indicate in which instances live vs headless females are used. Lines 594-596 are particularly confusing.

RESPONSE: We have now added this information in the material and methods section lines 600-602 and rewrote the lines 607-610.

Please comment on the size of the species used in the assays compared to the size of the chamber. The size of the chamber can have a drastic effect on behaviors^{3, 4}. Is a 1 cm diameter chamber sufficient for each of the different species tested?

RESPONSE: Single-pair courtships assays were performed in a chamber (1 cm diameter × 0.5 cm depth), while the competition courtship experiments (competitive experiments with 2 males and 1 female; competitive experiments with 2 females and 1 male) were performed in a chamber (5 cm diameter × 1 cm depth). We have now added this in the material and methods lines 600-602 and 607-610.

1. Mori, K. (2007) Significance of chirality in pheromone science, *Bioorg. Med. Chem.* 15, 7505-7523.
2. Joseph, R. M., and Carlson, J. R. (2015) *Drosophila* Chemoreceptors: A Molecular Interface Between the Chemical World and the Brain, *Trends Genet* 31, 683-695.
3. Kravitz, E. A., and Fernandez, M. P. (2015) Aggression in *Drosophila*, *Behav Neurosci* 129, 549-563.
4. Griffith, L. C., and Ejima, A. (2009) Courtship learning in *Drosophila melanogaster*: diverse plasticity of a reproductive behavior, *Learning & memory* (Cold Spring Harbor, N.Y.) 16, 743-750.

References:

- Bartelt, R.J., Jackson, L.L., and Schaner, A.M. (1985). Ester Components of Aggregation Pheromone of *Drosophila-Virilis* (Diptera, Drosophilidae). *J Chem Ecol* 11, 1197-1208.
- Dweck, H.K.M., Ebrahim, S.A.M., Thoma, M., Mohamed, A.A.M., Keeseey, I.W., Trona, F., Lavista-Llanos, S., Svatos, A., Sachse, S., Knaden, M., *et al.* (2015). Pheromones mediating copulation and attraction in *Drosophila*. *P Natl Acad Sci USA* 112, E2829-E2835.
- Huan, T., Forsberg, E.M., Rinehart, D., Johnson, C.H., Ivanisevic, J., Benton, H.P., Fang, M., Aisporna, A., Hilmers, B., Poole, F.L., *et al.* (2017). Systems biology guided by XCMS Online metabolomics. *Nat Methods* 14, 461-462.
- Khallaf, M.A., Auer, T.O., Grabe, V., Depetris-Chauvin, A., Ammagarahalli, B., Zhang, D.-D., Lavista-Llanos, S., Kaftan, F., Weißflog, J., Matzkin, L.M., *et al.* (2020). Mate discrimination among subspecies through a conserved olfactory pathway. *Sci Adv* 6, eaba5279.
- Lin, H.H., Cao, D.S., Sethi, S., Zeng, Z., Chin, J.S.R., Chakraborty, T.S., Shepherd, A.K., Nguyen, C.A., Yew, J.Y., Su, C.Y., *et al.* (2016). Hormonal modulation of pheromone detection enhances male courtship success. *Neuron* 90, 1272-1285.
- Obbard, D.J., Maclennan, J., Kim, K.W., Rambaut, A., O'Grady, P.M., and Jiggins, F.M. (2012). Estimating divergence dates and substitution rates in the *Drosophila* phylogeny. *Mol Biol Evol* 29, 3459-3473.
- Revadi, S., Lebreton, S., Witzgall, P., Anfora, G., Dekker, T., and Becher, P.G. (2015). Sexual Behavior of *Drosophila suzukii*. *Insects* 6, 183-196.
- Russo, C.A.M., Mello, B., Frazão, A., and Voloch, C.M. (2013). Phylogenetic analysis and a time tree for a large drosophilid data set (Diptera: Drosophilidae). *Zoological Journal of the Linnean Society* 169, 765-775.
- Snellings, Y., Herrera, B., Wildemann, B., Beelen, M., Zwarts, L., Wenseleers, T., and Callaerts, P. (2018). The role of cuticular hydrocarbons in mate recognition in *Drosophila suzukii*. *Sci Rep-Uk* 8.

REVIEWERS' COMMENTS

Reviewer #1 (Remarks to the Author):

The authors addressed my concerns in their revised manuscript. I have no remaining concerns.

This is an impressive body of work. Congratulations!

Chris Potter
Johns Hopkins University

Reviewer #3 (Remarks to the Author):

I thank the authors for carefully addressing my concerns, including additional interesting analyses about heterospecific compound detection, and making all data types available for general access. The analyses from this massive and substantial manuscript will greatly benefit the field of chemical communication and provide a wealth of data that will be mined for many years to come.

I have a few minor corrections:

Line 85: use of “host” is confusing - suggest “plant substrate” or “host plant” rather than “host”

106: remove “always”

195: remove “in addition”

204: sentence discusses male-specific compounds (Fig 2b) but Supp Fig 2b shows female compounds

269: remove “an” before “evidence”

332: Please cite the following as it also shows the use of transferred pheromones as a mate-guarding strategy in *D. mojavensis* and *D. arizonae*: Chin JS, Ellis SR, Pham HT, Blanksby SJ, Mori K, Koh QL, Etges WJ, Yew JY. Sex-specific triacylglycerides are widely conserved in *Drosophila* and mediate mating behavior. *Elife*. 2014 Mar 11;3:e01751. doi: 10.7554/eLife.01751. PMID: 24618898; PMCID: PMC3948109.

341: change “have” to “has”

355: change “All these females” to “Each of these females”

382-386: This section is confusing – the first sentence states that “males have significantly more

chemicals with higher phylogenetic signals than females” while the next sentence states that “many of the male-specific compounds display low phylogenetic signals”. Does this mean that even though male compounds have low phylogenetic signal, the signal is still higher than with females? Please re-write for clarity.

390: Please clarify what is meant by “aggregation pheromones in beetles – used as sexual signals”. Aggregation pheromones act on both sexes as an attractant while sexual signals act on one sex.

422: remove “all the”

460 – 466: This section is very confusing and should be re-written for grammar and clarity. What does “central level” refer to – central brain circuits? Proving this point would require measurements of central circuit activity. Please restate in more speculative terms. It would be sufficient to say that a change in valence is likely encoded at the level of central circuits.

Supp Fig 4C: change DCM in legend to hexane.

***Nature communications* (NCOMMS-20-44553A): Response to reviewers' comments**

RESPONSE: We thank the editor and the reviewers for their careful reading of our manuscript and for their positive comments.

Reviewer #1 (Remarks to the Author):

The authors addressed my concerns in their revised manuscript. I have no remaining concerns.

This is an impressive body of work. Congratulations!

Chris Potter

Johns Hopkins University

RESPONSE: We thank Prof. Dr. Christopher Potter for these extremely positive comments.

Reviewer #3 (Remarks to the Author):

I thank the authors for carefully addressing my concerns, including additional interesting analyses about heterospecific compound detection, and making all data types available for general access. The analyses from this massive and substantial manuscript will greatly benefit the field of chemical communication and provide a wealth of data that will be mined for many years to come.

RESPONSE: Thank you very much for these positive and very constructive comments!

I have a few minor corrections:

Line 85: use of "host" is confusing - suggest "plant substrate" or "host plant" rather than "host"

RESPONSE: We have now changed "host" to "host plant".

106: remove "always"

RESPONSE: We have now removed "always".

195: remove "in addition"

RESPONSE: We have now removed "in addition".

204: sentence discusses male-specific compounds (Fig 2b) but Supp Fig 2b shows female compounds

RESPONSE: We have now removed "Supplementary Figure 2B".

269: remove "an" before "evidence"

RESPONSE: We have now removed "an".

332: Please cite the following as it also shows the use of transferred pheromones as a mate-guarding strategy in *D. mojavensis* and *D. arizonae*: Chin JS, Ellis SR, Pham HT, Blanksby SJ, Mori K, Koh QL, Etges WJ, Yew JY. Sex-specific triacylglycerides are widely conserved in *Drosophila* and mediate mating behavior. *Elife*. 2014 Mar 11;3:e01751. doi: 10.7554/eLife.01751. PMID: 24618898; PMCID: PMC3948109.

RESPONSE: We apologize for the oversight and now added the reference.

341: change “have” to “has”

RESPONSE: We have now changed “have” to “has”.

355: change “All these females” to “Each of these females”

RESPONSE: We have now changed “All these females” to “Each of these females”.

382-386: This section is confusing – the first sentence states that “males have significantly more chemicals with higher phylogenetic signals than females” while the next sentence states that “many of the male-specific compounds display low phylogenetic signals”. Does this mean that even though male compounds have low phylogenetic signal, the signal is still higher than with females? Please re-write for clarity.

RESPONSE: We thank the reviewer for her/his comment. We have rewritten the sentences to “Our findings reveal that many of the male-specific compounds display low phylogenetic signals (i.e., less conserved in the closely related species) (Fig. 2b; Supplementary Figure 2B, C), which might result in divergence of sexual communication among the sibling species. However, compared to females, males have significantly more chemicals with higher phylogenetic signals, i.e. there is a better correlation between genetic and chemical distance in males (Supplementary Figure 1C).”

390: Please clarify what is meant by “aggregation pheromones in beetles – used as sexual signals”.

Aggregation pheromones act on both sexes as an attractant while sexual signals act on one sex.

RESPONSE: We have now deleted “ – used as sexual signals – ”.

422: remove “all the”

RESPONSE: We have now deleted “all the”.

460 – 466: This section is very confusing and should be re-written for grammar and clarity. What does “central level” refer to – central brain circuits? Proving this point would require measurements of central circuit activity. Please restate in more speculative terms. It would be sufficient to say that a change in valence is likely encoded at the level of central circuits.

RESPONSE: We have rewritten the sentence to be “These results reveal that species retain – at the peripheral level – the ability to detect the chemicals no longer produced by conspecifics, but a change in valence is likely encoded at the level of central circuits⁹⁵.”

Supp Fig 4C: change DCM in legend to hexane.

RESPONSE: We have changed “DCM” to “hexane”.